# Bayes with No Shame:
# Admissibility Geometries of Predictive Inference

## Abstract

Modern predictive systems compose several inferential modules—a predictor, a sequential monitor, a prediction set, an online strategy—each carrying its own certificate of optimality, and these certificates are often transported informally from one module to the next. We ask when such transport is licensed. Four admissibility geometries shape sequential and distribution-free inference: Blackwell risk dominance over convex risk sets, anytime-valid admissibility within the nonnegative supermartingale cone, fixed-level marginal coverage with expected-length efficiency on the exchangeable prediction-set frontier, and choice-based approachability (CApp) boundary-feasibility, under which each parameter's time-averaged risk reaches its oracle boundary value in Cesàro average. Each geometry carries a distinct certificate of optimality: a supporting-hyperplane prior, a nonnegative supermartingale, a conformal exchangeability rank, and a Cesàro steering argument, respectively. We embed all four in a common ambient space, the space $\Sigma$ of *predictive systems*: tuples $\Delta = (\delta, E, \hat{C}, \sigma)$ of predictor, $e$-process, prediction set, and online strategy. Within $\Sigma$ we prove a *witness-based non-nesting theorem*: for each ordered pair of the four criterion classes, an explicit predictive system, active in both relevant coordinates, lies in one class but not the other (Theorems 5.21 and 6.12). The theorem records non-nesting across categorically different object spaces and partial orders; it does not assert that the four criteria are practically incompatible, and any joint ranking requires an additional normative choice not supplied by the theories themselves.

We separate the martingale claims by provenance: proved in general, cited, or established by the Bernoulli log-loss counterexample. Bayesian posterior predictives under a single prior are martingales under their prior predictive law (general). Anytime-valid admissibility within $e$-processes is equivalent to the nonnegative martingale property (cited result). Self-consistency of a predictor under its own predictive law does not imply Blackwell admissibility (established by the Bernoulli log-loss counterexample, Theorem 4.4); coverage validity and CApp boundary-feasibility require no martingale structure.

The four criteria admit a common *design schema* (specify a criterion-specific feasibility constraint, then optimize Bayesian integrated risk within it), but the decision spaces, partial orders, and risk functionals differ across criteria. The schema organizes the four paradigms as a recipe; it does not reduce them to a single optimization problem. Admissibility is irreducibly criterion-relative.

**Keywords.** admissibility; Blackwell dominance; martingales; e-processes; conformal prediction; sequential inference; criterion separation.

**MSC 2020.** 62C15; 62F03; 62L10; 60G42.

## 1  Introduction

A classical example in Blackwell-style statistical decision theory (Blackwell & Girshick, 1954) considers predicting the next outcome of a Bernoulli sequence under log loss. The natural plug-in rule $\hat{p}_n = S_n/n$

looks beyond reproach: it is a martingale under its own predictive law, it concentrates on the true $\theta$ by the law of large numbers, and it is the maximum likelihood estimator at every sample size. Yet for every $\theta \in (0,1)$ and every $n \geq 1$ it is strictly dominated by the Bayes predictive $\hat{p}_n^B = (S_n + \frac{1}{2})/(n+1)$. The plug-in assigns probability zero to events that occur with positive probability, incurring infinite log-loss risk on the event $\{S_n \in \{0,n\}\}$; its martingale coherence does not rescue it from inadmissibility. Blackwell's point was geometric: the plug-in risk vector sits above the lower boundary of the convex risk set, and a supporting-hyperplane prior, the Jeffreys $\text{Beta}(\frac{1}{2}, \frac{1}{2})$, exposes a strictly dominating Bayes alternative on that boundary.

In modern predictive systems the same phenomenon propagates, but with new actors and new certificates of optimality. Bayesian predictive modeling certifies optimality through proper scoring rules and posterior predictives; conformal prediction (Vovk et al., 2005a) achieves distribution-free coverage without optimizing any loss; $e$-processes and safe anytime-valid testing (Shafer & Vovk, 2019; Ramdas et al., 2023; Grünwald et al., 2024) control type-I error at every stopping time by a structural condition (the nonnegative martingale property) that has no analogue in classical risk theory; and online calibration and defensive forecasting (Foster & Vohra, 1998; Vovk et al., 2005b) achieve calibration in the Cesàro sense through fixed-point arguments, without optimizing any per-round loss function. Each method is "optimal" in its own sense, yet no single criterion governs all four. The paper is about that word "optimal": what it means, what certifies it, and why the certificate does not travel across inferential worlds.

**Practical motivation: certificate transport in modern AI systems.** Modern AI systems increasingly compose several inferential modules in a single pipeline. A foundation model may be trained under a proper scoring rule, monitored through sequential safety indicators, wrapped by a conformal uncertainty layer, and evaluated for calibration after deployment. Each module carries its own certificate of optimality, yet those certificates are often transported informally across components. A predictor that is Bayes-optimal under log loss is not automatically anytime-valid; a conformal guarantee does not imply Blackwell admissibility; calibration on a held-out stream does not imply risk optimality at any finite sample size. The practical question is therefore not merely whether a system is optimal, but optimal with respect to *which certificate*, and what guarantees it does *not* inherit from the other modules with which it composes. This paper is not a critique of any one methodology. Its claim is that these methods are increasingly combined in a single system, and that their certificates of optimality do not automatically transfer across components.

We use "no-shame" throughout as a mnemonic for Blackwell-admissible, and the term is chosen deliberately. Williams (1993) distinguishes guilt, which answers to an external rule, from shame, which is the recognition of falling below a standard one endorses oneself. Blackwell dominance is shame in precisely this sense: a dominated rule is condemned by no outside authority, only by the loss function the analyst herself declared. The metaphor does rhetorical work because it locates the criticism *inside* the criterion; for the same reason it points to pluralism rather than to a ranking. A verdict that is internal to a standard offers no standard-free court of appeal: the analyst who endorses anytime validity, or marginal coverage, or long-run calibration, answers to that standard and not automatically to the others. The paper's formal content is order-theoretic; the metaphor plays no role in the results, and readers may substitute "Blackwell-admissible" wherever "no-shame" appears. But the structure it gestures at is the structure the theorems prove: admissibility is criterion-relative, and the separation theorems (Theorems 5.21 and 6.12) make this precise.

The four criteria, each defined relative to a different performance objective and a different certificate of optimality, are as follows. *Blackwell admissibility* (Sections 2–3): no competing rule has uniformly lower risk over $\Theta$; certificate is a supporting-hyperplane prior. *Anytime-valid admissibility* (Section 5): the e-process is a nonnegative martingale under the null, equivalent to admissibility within $\mathcal{C}_{\text{AV}}$; certificate is a nonnegative supermartingale. *Marginal coverage validity* (Section 5): $\mathbb{P}(Y_{n+1} \in \hat{C}_n(X_{n+1})) \geq 1 - \alpha$ under exchangeability, with expected-length efficiency within the fixed-level feasible class; certificate is a conformal exchangeability rank. *CApp boundary-feasibility* (Section 6): the time-averaged risk converges to the oracle boundary value $\partial_- \mathcal{R}(\theta)$ for every $\theta$; certificate is a Cesàro steering argument.

Our contributions are as follows.

(1) *Witness-based non-nesting in a common space.* We introduce the space $\Sigma$ of predictive systems $\Delta = (\delta, E, \hat{C}, \sigma)$ (Definition 5.1) as a common ambient object for the four criteria, and prove that the four corresponding admissibility classes $\mathfrak{B}, \mathfrak{A}, \mathfrak{C}, \mathfrak{D} \subseteq \Sigma$ are pairwise non-nested by exhibiting, for every ordered pair of classes, an explicit predictive system that is active in both relevant coordinates and lies in one class but not the other (Theorems 5.21 and 6.12). This provides a formal language for reasoning about modular predictive systems whose components satisfy different inferential guarantees. The theorem is structural and substantive within $\Sigma$ but does not claim practical incompatibility outside of it; any joint ranking of the four criteria requires an additional normative choice not supplied by the four theories.

(2) *Distinguished martingale claims.* We separate three notions of coherence (Definition 4.1): fixed-sample Blackwell admissibility, single-prior sequential Bayes coherence, and self-consistency under a data-dependent predictive law. Single-prior Bayes implies martingale coherence under the prior predictive (Proposition 4.3), and within $\mathcal{C}_{\mathrm{AV}}$ the nonnegative martingale property is equivalent to admissibility (Ramdas et al., 2022). The Bernoulli log-loss counterexample (Theorem 4.4) supplies a self-consistent predictor that is not Blackwell admissible; this witness, not a general theorem, establishes that self-consistency is not sufficient for Blackwell admissibility.

(3) *Constrained Bayes as a design schema.* We introduce a constrained Bayes schema (Definition 5.18) under which each criterion specifies a criterion-specific decision space $\mathcal{D}_C$, feasibility set $\mathcal{F}_C \subseteq \mathcal{D}_C$, and risk functional $\mathcal{R}_C$. This is a recipe rather than a unifying theorem (Remark 5.19); the four resulting optimization problems live on different decision spaces and remain non-nested.

A natural objection is that the separation is trivially definitional: Bayes prediction, e-processes, conformal sets, and defensive forecasters "solve different problems," so of course they are not nested. The content of the separation theorems is that the non-nesting persists even when all four frameworks are applied to the *same* statistical experiment (the Bernoulli model with a common parameter $\theta$): each framework imposes a different partial order on procedures derived from that process, and none of the four orders subsumes the others. Any joint ranking requires an additional normative choice not supplied by the four theories themselves.

In short: Bayes reaches $\partial_- \mathcal{R}$ by supporting hyperplanes; anytime validity by nonnegative martingales; conformal prediction by exchangeability ranks; and approachability by Cesàro steering. These are different certificates of admissibility, and they certify different objects.

**Roadmap.** Sections 2–3 develop the Blackwell geometry on the convex risk set, including the corrected supporting-hyperplane identification and the full-support Bayes-implies-no-shame corollaries. Section 4 introduces the martingale layer and the three coherence notions, and establishes the plug-in counterexample. Section 5 introduces the predictive-system meta-space $\Sigma$, defines fixed-level coverage admissibility, presents the constrained Bayes schema, and proves the first separation theorem (Theorem 5.21). Section 6 develops the constructive–Cesàro distinction and the extended separation (Theorem 6.12) including the misspecified-prior witness. Section 7 collects the Bernoulli laboratory and Section 8 reports three Monte Carlo illustrations. Section 9 reframes forecasting, sequential testing, and conformal wrappers as motivations rather than consequences; Section 10 closes with the discussion.

## 2  Primitive Objects

The decision-theoretic framework requires five objects: a parameter space, an action space, a loss function, a sample space, and a statistical model. We adopt the extended-real formulation that allows $+\infty$ risk, accommodating proper scoring rules such as log loss from the outset.

**Definition 2.1** (Statistical decision problem)**.** A *statistical decision problem* is a tuple $(\Theta, \mathcal{A}, L, \mathcal{X}, \mathcal{P})$ where:

    (i) $\Theta \subset \mathbb{R}^d$ is the *parameter space*, compact and metrizable.
    (ii) $\mathcal{A} \subset \mathbb{R}^m$ is the *action space*, compact and metrizable.
    (iii) $\mathcal{X}$ is the *sample space*, a Polish space, and $\mathcal{P} = \{P_\theta : \theta \in \Theta\}$ is the statistical model.
    (iv) $L : \Theta \times \mathcal{A} \to [0, \infty]$ is the *loss function*, satisfying:
        (a) $L(\theta, a)$ is measurable in $\theta$ for every $a$;

(b) $L(\theta, \cdot)$ is lower semicontinuous for every $\theta$;

(c) $L$ is bounded below (by zero, without loss of generality).

Risks are allowed to take the value $+\infty$; dominance is defined in the extended-real sense via the coordinatewise ordering on $[0, \infty]^\Theta$.

**Definition 2.2** (Decision rules)**.** Given data $X^n = (X_1, \ldots, X_n) \in \mathcal{X}^n$, a *(randomized) decision rule* is a measurable map $\delta : \mathcal{X}^n \to \Delta(\mathcal{A})$, where $\Delta(\mathcal{A})$ denotes the probability measures on $\mathcal{A}$. The class $\mathcal{D}$ of all decision rules is convex: for $\lambda \in [0, 1]$, the mixture $\delta_\lambda = \lambda\delta_1 + (1 - \lambda)\delta_2$ is defined by drawing from $\delta_1$ or $\delta_2$ with probabilities $\lambda$ and $1 - \lambda$ independently of $X^n$.

**Why extended-real risk.** Proper scoring rules (Gneiting & Raftery, 2007) used in this paper, chiefly the log loss $L(\theta, p) = -\theta \log p - (1 - \theta) \log(1 - p)$, take the value $+\infty$ at boundary predictions $p \in \{0, 1\}$ when $\theta \in (0, 1)$. Allowing $R(\theta, \delta) \in [0, \infty]$ from the outset is therefore essential, not a technicality: the plug-in counterexample of Section 4 hinges on a predictor whose risk is $+\infty$ with positive probability, and truncating to bounded loss would erase the phenomenon.

**Definition 2.3** (Risk function)**.** The *risk function* of a (randomized) decision rule $\delta \in \mathcal{D}$ is

$$R(\theta, \delta) \;=\; \mathbb{E}_\theta\left[ \int_\mathcal{A} L(\theta, a) \, \mathrm{d}\delta(X^n)(a) \right] \;\in\; [0, \infty], \qquad \theta \in \Theta,$$

where the inner integral marginalizes over the action $a$ drawn from the conditional distribution $\delta(X^n) \in \Delta(\mathcal{A})$, and the outer expectation is over $X^n \sim P_\theta$. For deterministic rules $\delta(X^n) = a(X^n)$ this reduces to $R(\theta, \delta) = \mathbb{E}_\theta[L(\theta, a(X^n))]$. Bayes risks are well-defined in $[0, \infty]$. Convexity follows from randomization. Lower semicontinuity under the topology used in Section 3 is part of the standing regularity assumptions stated there, and follows by Fatou's lemma in the finite-sample settings used for the explicit witnesses.

**Example 2.4** (Bernoulli log-loss, our running illustration)**.** Throughout the paper we will return to the following concrete instance of Definitions 2.1–2.3. Let $X_i \overset{\text{iid}}{\sim} \text{Bernoulli}(\theta)$ with $\theta \in \Theta \subseteq (0, 1)$. The action space is $\mathcal{A} = [0, 1]$ (a forecast probability). The loss is the binary log loss

$$L(\theta, p) = -\theta \log p - (1 - \theta) \log(1 - p), \qquad p \in [0, 1],$$

extended by $L(\theta, 0) = L(\theta, 1) = +\infty$ when $\theta \in (0, 1)$. For a deterministic predictor $\hat{p}_n = \hat{p}_n(X^n)$, the risk is

$$R(\theta, \hat{p}_n) = \mathbb{E}_\theta\big[L(\theta, \hat{p}_n(X^n))\big] \;=\; \mathbb{E}_\theta\big[D_{\mathrm{KL}}\big(\text{Bern}(\theta) \, \| \, \text{Bern}(\hat{p}_n(X^n))\big)\big] + H(\theta),$$

where $H(\theta) = -\theta \log \theta - (1 - \theta) \log(1 - \theta)$ is the binary entropy and $D_{\mathrm{KL}}$ is the Kullback–Leibler divergence. The entropy $H(\theta)$ is the *oracle coordinatewise lower envelope* under log loss: it is the risk that would be attained at $\theta$ by the unattainable rule $\hat{p}_n \equiv \theta$. The excess risk of any data-based predictor is the expected $D_{\mathrm{KL}}$ term in the display above, which is nonnegative and, for the shrinkage and universal-coding predictors considered below, vanishes asymptotically. Fixed-sample Blackwell admissibility (Definition 2.5 below and Section 3) is coordinatewise undominatedness of the full risk vector across $\Theta$, not pointwise attainment of $H(\theta)$. When compactness is needed for complete-class arguments we work on compact Bernoulli submodels $\Theta = [\eta, 1 - \eta] \subset (0, 1)$; the plug-in pathology itself holds pointwise for every $\theta \in (0, 1)$. In the Cesàro/log-loss examples of Section 6, universal coding strategies such as KT (Krichevsky & Trofimov, 1981) drive the average regret against the best constant in hindsight to zero, so their time-averaged risk approaches the oracle envelope $H(\theta)$ at rate $O((\log n)/n)$. We use this Bernoulli example as the running illustration for every notion in the paper (predictor, $e$-process, prediction set, online strategy).

**Definition 2.5** (Dominance and admissibility)**.** The partial order on $[0, \infty]^\Theta$ is defined by $r \leq r'$ if and only if $r(\theta) \leq r'(\theta)$ for all $\theta \in \Theta$. Rule $\delta'$ *dominates* $\delta$ if $R(\theta, \delta') \leq R(\theta, \delta)$ for all $\theta \in \Theta$ with strict inequality for some $\theta_0 \in \Theta$. A rule $\delta$ is *Blackwell admissible* if no rule in $\mathcal{D}$ dominates it.

**Definition 2.6** (Risk set)**.** For $\Theta = \{\theta_1, \ldots, \theta_k\}$ finite, associate to each $\delta \in \mathcal{D}$ its *risk vector* $r(\delta) = \big(R(\theta_1, \delta), \ldots, R(\theta_k, \delta)\big) \in [0, \infty]^k$. The *risk set* is

$$\mathcal{R} \;=\; \big\{r(\delta) : \delta \in \mathcal{D}\big\} \;\subset\; [0, \infty]^k.$$

The risk set is the image of the decision space under the risk map; its geometry encodes which rules dominate which. The finite-$\Theta$ representation is the primary geometric display; compact-$\Theta$ extensions are handled through the standard complete-class regularity assumptions of Section 3 (the *Standing regularity* hypotheses below).

## 3 Geometry of No-Shame

Admissibility has a geometric characterization: a rule is admissible if and only if its risk vector lies on the lower boundary of the risk set. This section establishes convexity, existence of Bayes rules, and closedness of the risk set, and shows that every admissible rule is supported by a prior, the geometric content of the no-shame principle.

**Standing regularity.** Throughout Section 3 we use the standard compactness and lower-semicontinuity hypotheses required for the Wald–Blackwell complete-class argument (Wald, 1950; Blackwell & Girshick, 1954; Berge, 1963). In the finite-sample and finite-action settings used by the explicit witnesses these conditions are automatic. In general Polish sample spaces with compact action sets, we assume compactness of the randomized-rule class $\mathcal{D}$ in the topology of pointwise weak convergence and lower semicontinuity of $\delta \mapsto \int_\Theta R(\theta, \delta)\, d\Pi(\theta)$ for finitely supported priors $\Pi$. Both conditions follow from Prokhorov's theorem applied to $\Delta(\mathcal{A})$ (Aliprantis & Border, 2006, Ch. 15) and Fatou's lemma, respectively.

### 3.1 Convexity of the risk set

**Lemma 3.1** (Convexity)**.** *Under Definitions 2.1–2.6, $\mathcal{R}$ is convex.*

*Proof.* Let $\delta_1, \delta_2 \in \mathcal{D}$ and $\lambda \in [0, 1]$. The mixture $\delta_\lambda$ satisfies $R(\theta_j, \delta_\lambda) = \lambda R(\theta_j, \delta_1) + (1 - \lambda)R(\theta_j, \delta_2)$ for each $j$, so $r(\delta_\lambda) = \lambda r(\delta_1) + (1 - \lambda)r(\delta_2) \in \mathcal{R}$. $\square$

### 3.2 Existence of Bayes rules

**Lemma 3.2** (Existence of Bayes rules)**.** *Under the standing regularity assumptions of Section 3, every prior $\Pi$ with finite integrated risk admits a Bayes rule $\delta_\Pi \in \arg\min_{\delta \in \mathcal{D}} \int_\Theta R(\theta, \delta)\, d\Pi(\theta)$.*

*Proof.* By the standing regularity assumption, $\mathcal{D}$ is compact in the topology under consideration and the integrated-risk functional $\delta \mapsto \int_\Theta R(\theta, \delta)\, d\Pi(\theta)$ is lower semicontinuous. A lower semicontinuous functional attains its minimum on a compact set. $\square$

### 3.3 Closedness of the lower-comprehensive hull

**Proposition 3.3** (Lower-comprehensive closedness)**.** *Under Definition 2.1, the lower-comprehensive hull*

$$\mathcal{R}_+ := \mathcal{R} + \mathbb{R}_+^k = \{\, r \in [0, \infty]^k : r \geq r(\delta) \text{ coordinatewise for some } \delta \in \mathcal{D} \,\}$$

*is closed in $[0, \infty]^k$ with the product topology. Equivalently, if $r(\delta_\alpha) \to r^*$ in $[0, \infty]^k$ along a net, then there exists $\delta^* \in \mathcal{D}$ with $r(\delta^*) \leq r^*$ coordinatewise. Since $\Theta$ is finite in this section, $[0, \infty]^k$ is compact Hausdorff in the product topology.*

*Proof.* Let $(r(\delta_\alpha))$ be a net in $\mathcal{R}$ with $r(\delta_\alpha) \to r^*$ in the product topology of $[0, \infty]^k$. Compactness of $\mathcal{D}$ (by the standing regularity assumptions) supplies a subnet with $\delta_\alpha \to \delta^*$ weakly. Lower semicontinuity of $R(\theta_j, \cdot)$ for each $j$ gives $R(\theta_j, \delta^*) \leq \liminf_\alpha R(\theta_j, \delta_\alpha) = r_j^*$ componentwise. Hence $r(\delta^*) \leq r^*$ coordinatewise, so $r^* \in r(\delta^*) + \mathbb{R}_+^k \subseteq \mathcal{R}_+$. Stability of $\mathcal{R}_+$ under further upward translation by $\mathbb{R}_+^k$ is immediate. $\square$

*Remark* 3.4 (Why $\mathcal{R}$ need not itself be closed). Lower semicontinuity of risk delivers a limit point $r(\delta^*) \leq r^*$, not $r(\delta^*) = r^*$. The hull $\mathcal{R}_+$ is the right closed object for the supporting-hyperplane argument (Theorem 3.8): boundary points of $\mathcal{R}$ are boundary points of $\mathcal{R}_+$ but the supporting normal is obtained from $\mathcal{R}_+$, where

translation by the nonnegative orthant is allowed. Continuity of the risk map (rather than mere lower semicontinuity) would suffice for closedness of $\mathcal{R}$ itself but is not assumed here.

*Remark* 3.5. Extended-real values do not break convex separation. The supporting hyperplane argument in Theorem 3.8 is applied to $\mathcal{R} \cap \mathbb{R}_+^k$, not to $[0, \infty]^k$ directly. Any admissible point $r^*$ has finite coordinates under the prior that supports it: if $R(\theta_j, \delta^*) = +\infty$ for some $j$ with $\Pi(\theta_j) > 0$, then the integrated Bayes risk would be infinite, which contradicts optimality whenever the model admits at least one finite-risk competitor. In the examples considered below, finite-risk competitors exist (in the Bernoulli model under log loss, any constant predictor $p \in (0, 1)$ has finite risk for every $\theta \in (0, 1)$, since $L(\theta, p)$ is then bounded above), and the supporting-hyperplane argument is applied on the finite-risk portion of $\mathcal{R}$. Hence separation occurs in $\mathbb{R}^k$, and the extension to $[0, \infty]^k$ does not obstruct the argument.

## 3.4 Lower boundary and admissibility

**Definition 3.6** (Lower boundary)**.** The *lower boundary* of $\mathcal{R}$ is

$$\partial_- \mathcal{R} \;=\; \{r \in \mathcal{R} : \nexists\, r' \in \mathcal{R},\; r' \le r \text{ coordinatewise with } r' \ne r\}.$$

**Proposition 3.7** (Boundary characterization)**.** *A rule $\delta$ is Blackwell admissible if and only if $r(\delta) \in \partial_- \mathcal{R}$.*

*Proof.* If $r(\delta) \notin \partial_- \mathcal{R}$, there exists $r' \in \mathcal{R}$ with $r' \le r(\delta)$ coordinatewise, $r' \ne r(\delta)$; the corresponding rule dominates $\delta$. Conversely, if $r(\delta) \in \partial_- \mathcal{R}$, no such $r'$ exists in $\mathcal{R}$. $\qquad\square$

Figure 1 displays the geometry for $|\Theta| = 2$; Figure 2 instantiates it for Bernoulli log-loss prediction, where the plug-in MLE sits off the boundary and two Bayes predictives occupy distinct exposed points.

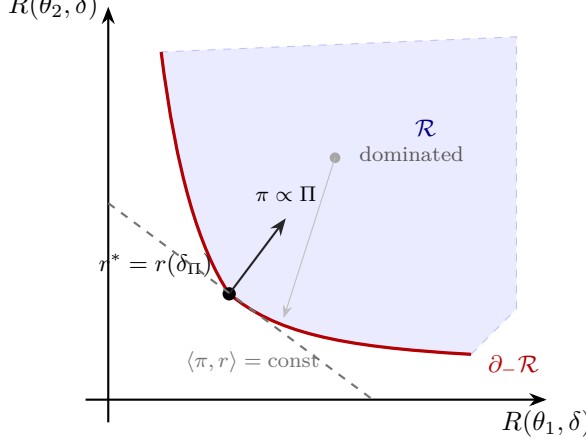

Figure 1: Risk set geometry for $|\Theta| = 2$. The convex risk set $\mathcal{R}$ (shaded) maps each decision rule to a risk vector. The lower boundary $\partial_- \mathcal{R}$ (bold curve) contains all admissible rules. At an admissible point $r^*$, the supporting hyperplane (dashed line) identifies the prior $\Pi$ whose normal $\pi$ defines the Bayes problem that $r^*$ solves (Theorem 3.8). Interior points are dominated.

## 3.5 Supporting hyperplanes and Bayes rules

**Notation convention ($\pi$ vs. $\Pi$).** Throughout the paper we maintain a strict notational separation between two related but distinct objects. The Greek letter $\pi$ (lower case) denotes the *geometric supporting-hyperplane normal vector* at a boundary point of $\mathcal{R}_+$; it lives in $\mathbb{R}_+^k \setminus \{0\}$. The capital $\Pi$ denotes the *normalized prior probability distribution* on $\Theta$ obtained by setting $\Pi = \pi / \|\pi\|_1$. Thus $\pi$ carries geometric content (it is the dual variable of a convex separation) while $\Pi$ carries probabilistic content (it integrates risk into Bayes risk). The two are equivalent up to normalization but are not interchangeable in the prose.

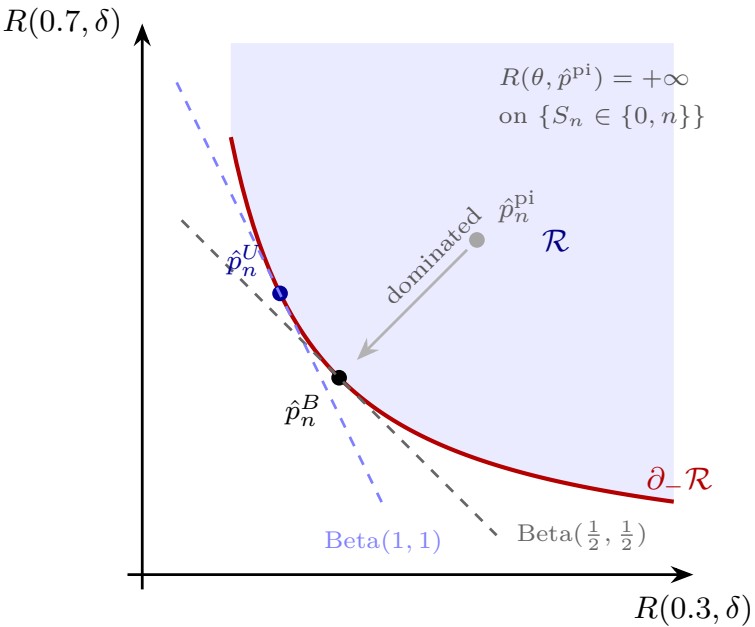

Figure 2: Concrete risk set for Bernoulli log-loss prediction with $\Theta = \{0.3, 0.7\}$, $n = 10$ (schematic). The Bayes predictive $\hat{p}_n^B = (S_n + \frac{1}{2})/(n + 1)$ under $\text{Beta}(\frac{1}{2}, \frac{1}{2})$ and the Laplace predictive $\hat{p}_n^U = (S_n + 1)/(n + 2)$ under $\text{Beta}(1, 1)$ both lie on the lower boundary $\partial_- \mathcal{R}$; each is no-shame with respect to its own prior, depicted by tangent (dashed) supporting hyperplanes. The plug-in MLE $\hat{p}_n^{\text{pi}} = S_n/n$ sits in the interior: it is dominated because it assigns probability zero to events that occur with positive probability, producing infinite log-loss contributions on $\{S_n \in \{0, n\}\}$. Its extended-real risk coordinates are $+\infty$ at every finite $n$; the dot marks a finite proxy position for visual comparison.

**Intuition for Theorem 3.8.** At any point on the lower boundary of the risk set, one can place a supporting hyperplane that touches the boundary at that point and lies weakly below every risk vector achievable by some decision rule. Convexity of $\mathcal{R}$ delivers the hyperplane; lower-comprehensiveness of $\mathcal{R}_+$ forces its normal to have nonnegative coordinates; nonnegativity lets us interpret the normal, after normalization, as a prior over $\Theta$. The theorem says that for every admissible risk vector there is such a prior, and that the admissible rule is then a Bayes rule under it.

**Theorem 3.8** (Supporting hyperplane identification). *Assume $|\Theta| = k$. If $r^* \in \partial_- \mathcal{R}$ and $\mathcal{R}$ is convex, there exists $\pi \in \mathbb{R}_+^k \setminus \{0\}$ such that*

$$\sum_{j=1}^k \pi_j R(\theta_j, \delta^*) \leq \sum_{j=1}^k \pi_j R(\theta_j, \delta) \qquad \text{for all } \delta \in \mathcal{D}.$$

*Setting $\Pi = \pi/\|\pi\|_1$ defines a prior on $\Theta$, and $\delta^*$ is a Bayes rule:*

$$\delta^* \in \arg\min_{\delta \in \mathcal{D}} \int R(\theta, \delta) \, d\Pi(\theta).$$

*Proof.* Consider the lower-comprehensive hull $\mathcal{R}_+ := \mathcal{R} + \mathbb{R}_+^k$. By Proposition 3.3, $\mathcal{R}_+$ is closed; by Lemma 3.1 it is convex. Since $r^* \in \partial_- \mathcal{R}$, no point $r \in \mathcal{R}$ satisfies $r \leq r^*$ with $r \neq r^*$, so $r^*$ lies on the boundary of $\mathcal{R}_+$. By the supporting-hyperplane theorem for closed convex sets (Rockafellar, 1970, Thm. 11.6), there exists $\pi \in (\mathbb{R}^k)^* \setminus \{0\}$ with

$$\langle \pi, r \rangle \geq \langle \pi, r^* \rangle \qquad \text{for all } r \in \mathcal{R}_+. \tag{1}$$

We show $\pi \geq 0$ coordinatewise. Fix $j \in \{1, \dots, k\}$ and $t > 0$. By lower-comprehensiveness of $\mathcal{R}_+$, $r^* + t e_j \in \mathcal{R}_+$ where $e_j$ is the $j$-th standard basis vector. Applying (1) at $r = r^* + t e_j$ yields $\langle \pi, r^* \rangle + t\pi_j \geq \langle \pi, r^* \rangle$, i.e.

$t\pi_j \geq 0$. Since $t > 0$ was arbitrary, $\pi_j \geq 0$. As $j$ was arbitrary, $\pi \in \mathbb{R}_+^k \setminus \{0\}$. Because $\mathcal{R} \subseteq \mathcal{R}_+$, (1) restricts to $\sum_j \pi_j R(\theta_j, \delta) \geq \sum_j \pi_j R(\theta_j, \delta^*)$ for every $\delta \in \mathcal{D}$. Normalizing $\Pi = \pi/\|\pi\|_1$ identifies $\delta^*$ with a Bayes minimizer under $\Pi$. □

*Remark* 3.9 (Scope of Theorem 3.8). The finite-$\Theta$ statement is the geometric display used in this paper. Compact-$\Theta$ versions require the standard complete-class regularity assumptions; Corollary 3.14 gives the support/continuity condition needed for the Bayes-implies-no-shame direction. See Blackwell & Girshick (1954) and Wald (1950) for the classical general development.

The converse direction (Bayes $\Rightarrow$ no-shame) requires support assumptions: a Bayes rule under a prior may be improved at a point of prior mass zero without changing the integrated risk under that prior, and lower semicontinuity alone does not exclude this. Corollaries 3.12 and 3.14 state the finite-$\Theta$ (full-support prior) and compact-$\Theta$ (full topological support, continuous risk) versions respectively; the general statement is the complete class theorem below.

**Theorem 3.10** (Wald–Blackwell complete class, under standard regularity (Wald, 1950; Blackwell & Girshick, 1954)). *Under Definition 2.1 together with the standing compactness, lower-semicontinuity, and closed-risk-set conditions of Section 3, every Blackwell admissible rule is a Bayes rule with respect to some prior $\Pi$ on $\Theta$, or a pointwise limit of Bayes rules. Equivalently, the class of Bayes rules is essentially complete.*

### 3.6 No-shame strategies

**Definition 3.11** (No-shame strategy). A rule $\delta \in \mathcal{D}$ is a *no-shame strategy* if it is Blackwell admissible; equivalently (by Proposition 3.7), if $r(\delta) \in \partial_- \mathcal{R}$.

**Corollary 3.12** (Full-support Bayes implies no-shame (finite $\Theta$)). *Let $\Theta = \{\theta_1, \ldots, \theta_k\}$ be finite and let $\Pi$ be a prior with $\Pi(\theta_j) > 0$ for every $j$. Every proper Bayes rule $\delta_\Pi$ with finite integrated risk under such a full-support prior is Blackwell admissible (no-shame).*

*Proof.* Suppose, for contradiction, that $\delta' \in \mathcal{D}$ dominates $\delta_\Pi$: $R(\theta_j, \delta') \leq R(\theta_j, \delta_\Pi)$ for all $j$, with strict inequality at some $j_0$. Multiplying by $\Pi(\theta_j) > 0$ and summing, $\sum_j \Pi(\theta_j) R(\theta_j, \delta') < \sum_j \Pi(\theta_j) R(\theta_j, \delta_\Pi)$, contradicting Bayes optimality of $\delta_\Pi$ under $\Pi$. □

*Remark* 3.13 (Direction of the equivalence). The converse of Corollary 3.12 requires care. Theorem 3.8 produces a supporting prior $\pi$ at every $r^* \in \partial_- \mathcal{R}$, but $\pi$ may assign zero mass to some coordinates. Hence every Blackwell admissible rule is Bayes *with respect to some prior*, possibly without full support, or is a pointwise limit of Bayes rules; this is the Wald–Blackwell complete class theorem (Theorem 3.10). We are therefore careful in the rest of the paper to distinguish the implication *full-support Bayes $\Rightarrow$ no-shame* from its converse. Admissibility under a non-full-support prior is witnessed directly by the risk-vector geometry (as in the point-mass example used in Theorem 6.12 (i)).

**Corollary 3.14** (Full-support Bayes implies no-shame (compact-$\Theta$ version)). *Let $\Theta$ be compact metrizable, $\Pi$ a Borel prior with full topological support on $\Theta$, and assume the risk function $\theta \mapsto R(\theta, \delta)$ is continuous for every $\delta \in \mathcal{D}$. Then every proper Bayes rule $\delta_\Pi$ with finite integrated risk is Blackwell admissible.*

*Proof.* Suppose, for contradiction, that $\delta'$ dominates $\delta_\Pi$. Then $R(\theta_0, \delta') < R(\theta_0, \delta_\Pi)$ at some $\theta_0 \in \Theta$, and $R(\theta, \delta') \leq R(\theta, \delta_\Pi)$ elsewhere. By continuity of $\theta \mapsto R(\theta, \delta') - R(\theta, \delta_\Pi)$, the strict inequality extends to an open neighborhood $U \ni \theta_0$. Full topological support of $\Pi$ gives $\Pi(U) > 0$, hence

$$\int_\Theta \big[ R(\theta, \delta_\Pi) - R(\theta, \delta') \big] \, \mathrm{d}\Pi(\theta) \geq \int_U \big[ R(\theta, \delta_\Pi) - R(\theta, \delta') \big] \, \mathrm{d}\Pi(\theta) > 0,$$

contradicting Bayes optimality of $\delta_\Pi$. □

*Remark* 3.15 (Why the split). The previous statement of this result implicitly required $\Pi$-positive mass at every $\theta$, which is the finite case; for compact $\Theta$ with general priors, a strict improvement at a single point $\theta_0$ with $\Pi(\{\theta_0\}) = 0$ does not automatically reduce the integrated Bayes risk. The compact-$\Theta$ statement adds the standard continuity-of-risk and full-support hypotheses, which suffice for the dominance to extend

to a neighborhood of positive $\Pi$-measure. For log loss on the Bernoulli model with $\Theta = (0, 1)$, continuity holds whenever the predictor avoids boundary predictions (cf. Theorem 4.4); the Bayes posterior predictives considered in this paper satisfy this requirement.

*Remark* 3.16 (Order-theoretic structure). Admissibility is defined by the coordinatewise partial order on $\mathbb{R}^k$, not by any metric. The supporting hyperplane in Theorem 3.8 is the linear-algebraic instrument for locating the prior $\Pi$ that rationalizes $\delta^*$; the dominance relation itself depends only on the order structure of $\mathbb{R}^k$. When validity constraints are imposed (anytime-valid error control or marginal coverage), the optimization in Theorem 3.8 restricts to a feasible subset of $\mathcal{D}$; the constrained Bayes formulation in Section 5.5 makes this precise.

*Remark* 3.17 (Structural scope of the admissibility results). The admissibility results of this section (existence of Bayes rules (Lemma 3.2), closedness and convexity of $\mathcal{R}$ (Lemma 3.1, Proposition 3.3), supporting hyperplane identification (Theorem 3.8), and the complete class theorem (Theorem 3.10)) rely on compactness and lower semicontinuity rather than bounded loss. This places the analysis within the classical Berge–Wald–Blackwell decision-theoretic framework (Berge, 1963; Wald, 1950; Blackwell & Girshick, 1954) and allows treatment of proper scoring rules such as log loss, which take the value $+\infty$ on the boundary of the probability simplex.

## 4 Martingale Layer

The risk-set geometry of Section 3 characterizes admissibility through priors and supporting hyperplanes. We now introduce a dynamic structure: the martingale property of Bayesian posterior predictive sequences, a property whose use in Bayesian asymptotics goes back to Doob (1949) and which underlies the recent martingale-posterior literature (Fong et al., 2023). This property characterizes single-prior sequential Bayes coherence and is central to anytime-valid inference, but it is neither necessary nor sufficient for fixed-sample Blackwell admissibility in general. The plug-in example of Section 4 shows in particular that self-consistency under a predictor's own predictive law does not imply Blackwell admissibility; the three coherence notions introduced next make these distinctions precise.

### 4.1 Three notions of coherence

The literature occasionally conflates several distinct "coherence" properties. We distinguish them explicitly here so that the plug-in pathology of Section 4 can be located precisely on the resulting map.

**Definition 4.1** (Three coherence notions). Let $(\delta_n)_{n \geq 1}$ be a sequence of decision rules on $(\mathcal{X}^n)_{n \geq 1}$.

(C1) *Fixed-sample Blackwell admissibility.* Each $\delta_n$ is Blackwell admissible in the sense of Definition 2.5: no $\delta' \in \mathcal{D}$ satisfies $R(\theta, \delta') \leq R(\theta, \delta_n)$ for all $\theta$ with strict inequality somewhere.

(C2) *Sequential Bayes coherence under $\Pi$.* There exists a single prior $\Pi$ such that, for every $n$, $\delta_n$ is the $n$-sample Bayes rule under $\Pi$; equivalently, the posterior predictive sequence $(m_n)$ of Definition 4.2 is a martingale under the prior predictive law $\tilde{P} = \int P_\theta \, d\Pi(\theta)$.

(C3) *Self-consistency under $\hat{P}$.* The sequence $(\delta_n)$ is a martingale under its own predictive law $\hat{P}$, in which $X_{n+1} \mid \mathcal{F}_n$ is distributed as $\delta_n$ itself.

The three notions are logically distinct. (C2) implies (C1) at each $n$ under the assumptions of Corollary 3.12 or 3.14. The implication (C3)⇒(C1) fails: the plug-in MLE $\hat{p}_n^{\mathrm{pi}} = S_n/n$ satisfies (C3) but is strictly dominated under log loss for every $\theta \in (0, 1)$ and every $n \geq 1$ (Theorem 4.4). Likewise (C1) does not imply (C2): a Blackwell-admissible rule may be Bayes under a different prior at each $n$, in which case the sequence is not coherent under any single $\Pi$. Proposition 4.3 below is the precise statement that (C2) holds: the martingale claim is under the *prior* predictive law $\tilde{P}$, not under any data-dependent predictive law $\hat{P}$.

Notion (C3) is the self-audit property at the heart of the well-calibrated Bayesian of Dawid (1982): a forecaster assessed under her own predictive law finds nothing to correct. The plug-in pathology of Theorem 4.4 is exactly the failure of (C3)⇒(C1): self-consistency under the predictor's own law is not a substitute for

non-dominance under the true data-generating process $P_\theta$. The Bayes corrections that resolve the pathology (e.g. $\mathrm{Beta}(\frac{1}{2}, \frac{1}{2})$ shrinkage) restore (C2), which in turn delivers (C1) by Corollary 3.12.

**Definition 4.2** (Posterior predictive sequence). Let $\Pi$ be a prior on $\Theta$ with $\mathcal{F}_n = \sigma(X_1, \ldots, X_n)$. In a general model, the posterior predictive is a probability kernel; equivalently, for every bounded measurable test function $h$, the process $m_n(h) = \mathbb{E}_\Pi[\mathbb{E}_\theta h(X_{n+1}) \,|\, \mathcal{F}_n]$ is a scalar martingale under the prior predictive law $\tilde{P} = \int P_\theta \, d\Pi(\theta)$. In the Bernoulli model, taking $h(x) = x$ recovers the *posterior predictive sequence*

$$m_n \;=\; \mathbb{E}_\Pi[\theta \,|\, \mathcal{F}_n], \qquad n \geq 0,$$

with $m_0 = \mathbb{E}_\Pi[\theta]$, which is the scalar version used throughout the paper.

**Proposition 4.3** (Bayes implies martingale). *Under Definition 4.2, $(m_n)_{n\geq 0}$ is a martingale with respect to $(\mathcal{F}_n)$ under the prior predictive measure $\tilde{P} = \int P_\theta \, d\Pi(\theta)$.*

*Proof.* By the tower property under $\tilde{P}$: $\mathbb{E}_{\tilde{P}}[m_n \,|\, \mathcal{F}_{n-1}] = \mathbb{E}_{\tilde{P}}[\mathbb{E}_\Pi[\theta \,|\, \mathcal{F}_n] \,|\, \mathcal{F}_{n-1}] = \mathbb{E}_\Pi[\theta \,|\, \mathcal{F}_{n-1}] = m_{n-1}$ a.s. Integrability holds since $\theta$ is bounded on compact $\Theta$. $\square$

## 4.2 Self-consistency is not sufficient for admissibility

**Theorem 4.4** (Self-consistency does not imply Blackwell admissibility). *In the Bernoulli model $X_i \overset{\text{iid}}{\sim} \mathrm{Bern}(\theta)$, $\theta \in (0,1)$, under log loss $L(\theta, p) = -\theta \log p - (1-\theta)\log(1-p)$:*

  *(i) Every Bayesian posterior predictive sequence $(m_n)$ is a martingale under the prior predictive measure (Proposition 4.3).*
  *(ii) The plug-in rule $\hat{p}_n^{\mathrm{pi}} = S_n/n$ satisfies the martingale condition under its own predictive measure $\hat{P}$ (where $X_t \,|\, X_{1:t-1} \sim \mathrm{Bern}(\hat{p}_{t-1}^{\mathrm{pi}})$) for $n \geq 2$, with any fixed initialization $\hat{p}_0 \in (0,1)$.*
  *(iii) $\hat{p}_n^{\mathrm{pi}}$ is strictly dominated by the Bayes rule $\hat{p}_n^B = (S_n + \frac{1}{2})/(n+1)$ under $\mathrm{Beta}(\frac{1}{2}, \frac{1}{2})$ prior, for every $n \geq 1$ and $\theta \in (0,1)$.*

*Hence $r(\hat{p}_n^{\mathrm{pi}}) \notin \partial_- \mathcal{R}$: self-consistency of a predictor under its own predictive law is not sufficient for Blackwell admissibility.*

*Proof. (i)* Proposition 4.3.

*(ii)* Under $\hat{P}$, $\mathbb{E}_{\hat{P}}[\hat{p}_n^{\mathrm{pi}} \,|\, X_{1:n-1}] = (S_{n-1} + \hat{p}_{n-1}^{\mathrm{pi}})/n = (S_{n-1} + S_{n-1}/(n-1))/n = S_{n-1}/(n-1) = \hat{p}_{n-1}^{\mathrm{pi}}$ a.s.

*(iii)* Under log loss, the risk decomposes as

$$R(\theta, \hat{p}_n) = \mathbb{E}_\theta\big[D_{\mathrm{KL}}(\mathrm{Bern}(\theta)\|\hat{p}_n)\big] + H(\theta),$$

where $H(\theta) = -\theta \log\theta - (1-\theta)\log(1-\theta)$ is the binary entropy. Hence the excess risk is

$$R(\theta, \hat{p}_n^{\mathrm{pi}}) - R(\theta, \hat{p}_n^B) = \mathbb{E}_\theta\big[D_{\mathrm{KL}}(\mathrm{Bern}(\theta)\|\hat{p}_n^{\mathrm{pi}}) - D_{\mathrm{KL}}(\mathrm{Bern}(\theta)\|\hat{p}_n^B)\big].$$

Since $\hat{p}_n^{\mathrm{pi}} = S_n/n \in \{0, \frac{1}{n}, \ldots, 1\}$ and $P_\theta(S_n = 0) = (1-\theta)^n > 0$ for every $\theta \in (0,1)$, the predictor $\hat{p}_n^{\mathrm{pi}} = 0$ assigns probability zero to $X_{n+1} = 1$, an event with probability $\theta > 0$; thus $D_{\mathrm{KL}}(\mathrm{Bern}(\theta)\|\hat{p}_n^{\mathrm{pi}}) = +\infty$ on the event $\{S_n = 0\}$ and $R(\theta, \hat{p}_n^{\mathrm{pi}}) = +\infty$. Meanwhile $\hat{p}_n^B = (S_n + \frac{1}{2})/(n+1) \in (0,1)$ for all $S_n \in \{0, \ldots, n\}$, so $R(\theta, \hat{p}_n^B) < \infty$. Hence $\hat{p}_n^B$ strictly dominates $\hat{p}_n^{\mathrm{pi}}$ for all $\theta \in (0,1)$ and $n \geq 1$, giving $r(\hat{p}_n^{\mathrm{pi}}) \notin \partial_- \mathcal{R}$ by Proposition 3.7. By Definition 3.11, $\hat{p}_n^{\mathrm{pi}}$ is not no-shame. $\square$

The failure is not that the plug-in lacks a martingale law; it is that the martingale law is the wrong one for Blackwell risk. The prior predictive law $\tilde{P}$, the plug-in self-predictive law $\hat{P}$, and the true model law $P_\theta$ are three distinct measures on the same path space; only the first two make the respective Bayes / plug-in sequences martingales, and only the third governs admissibility.

*Remark* 4.5 (Role of extended-real risk). The dominance in part (iii) requires $+\infty$ risk (Definitions 2.1 and 2.3). The boundary pathology ($D_{\mathrm{KL}}(\mathrm{Bern}(\theta)\|\hat{p}_n^{\mathrm{pi}}) = +\infty$ on $\{S_n = 0\}$) is eliminated if loss is bounded, but bounded losses exclude proper scoring rules such as log loss. Extended-real risk is therefore essential, not a technicality.

# 5 Criterion Separation

Sections 3–4 established Blackwell admissibility as the first geometry. We now introduce two additional admissibility criteria, anytime-valid sequential inference and marginal coverage validity, each operating on a different space of procedures with a different partial order, and prove that the three resulting classes are pairwise non-nested.

A natural objection is that predictors, $e$-processes, prediction sets, and online strategies are different kinds of mathematical objects. Meaningful non-nesting claims therefore require a common ambient product space in which each criterion attaches to its own coordinate. We construct that space, the space $\Sigma$ of *predictive systems*, before formulating any separation theorem.

## 5.1 The predictive-systems meta-space $\Sigma$

**Definition 5.1** (Predictive system)**.** Let $\mathcal{X}$ be a Polish single-observation space and let $\Omega = \mathcal{X}^\infty$ be the canonical path space, equipped with the product $\sigma$-algebra $\mathcal{F}_\infty$ and the natural filtration $\mathcal{F}_n = \sigma(X_1, \ldots, X_n)$. Let $\mathcal{P} = \{P_\theta : \theta \in \Theta\}$ be a statistical model on $(\Omega, \mathcal{F}_\infty)$, with $\Theta \subset \mathbb{R}^d$ compact and metrizable. A *predictive system* on this filtered space is a quadruple

$$\Delta = (\delta, E, \hat{C}, \sigma)$$

whose four components are $(\mathcal{F}_n)$-adapted processes:

(a) $\delta = (\delta_n)_{n \geq 1}$, where each $\delta_n$ is $\mathcal{F}_n$-measurable, taking values in $\Delta(\mathcal{A})$ with $\mathcal{A}$ compact metric and $\Delta(\mathcal{A})$ equipped with the weak topology (*predictor component*);
(b) $E = (E_n)_{n \geq 0}$, where $E_0 \equiv 1$, $E_n \geq 0$, and each $E_n$ is $\mathcal{F}_n$-measurable (*test component*);
(c) $\hat{C} = (\hat{C}_n)_{n \geq 0}$, where each $\hat{C}_n$ is an $\mathcal{F}_n$-measurable prediction-set process (*set component*);
(d) $\sigma = (\sigma_n)_{n \geq 1}$, where each $\sigma_n$ is $\mathcal{F}_{n-1}$-measurable, taking values in $\mathcal{A}$ (*strategy component*).

The collection of all such quadruples is denoted $\Sigma$. Fix once and for all a *default inactive system* $\Delta^\varnothing = (\delta^\varnothing, E^\varnothing, \hat{C}^\varnothing, \sigma^\varnothing)$, where $\delta_n^\varnothing \equiv a_\varnothing$ and $\sigma_n^\varnothing \equiv a_\varnothing$ for a fixed default action $a_\varnothing \in \mathcal{A}$, $E_n^\varnothing \equiv 1$, and $\hat{C}_n^\varnothing \equiv \mathcal{Y}$. A coordinate of $\Delta$ is *active* if it differs from the corresponding default coordinate on a set of positive $P_\theta$-probability for some $\theta \in \Theta$. In the Bernoulli illustrations below we take the default inactive action to be $a_\varnothing = \frac{1}{2}$, so the constant-zero predictor and constant-zero strategy used as failure witnesses in Theorem 5.21 are active coordinates.

**Standard-Borel character of $\Sigma$.** The space $\Sigma$ is defined entirely within ordinary Polish/Borel probability theory: all components are Borel-adapted processes on the canonical filtered path space. Every witness constructed in Theorems 5.21 and 6.12—the Bayes predictors, the plug-in rule, the likelihood-ratio $e$-process, the conformal threshold sets, the KT strategy, and the constant comparison rules—is given by an explicit measurable formula. Thus the meta-space $\Sigma$ is a bookkeeping device for typed predictive systems, not an additional compactification or nonstandard existence principle.

**Definition 5.2** (Criterion-relative admissibility on $\Sigma$)**.** For a predictive system $\Delta = (\delta, E, \hat{C}, \sigma) \in \Sigma$:

- $\Delta \in \mathfrak{B}$ iff $\delta$ is Blackwell admissible per Definition 2.5;
- $\Delta \in \mathfrak{A}$ iff $E \in \mathcal{C}_{\mathrm{AV}}$ (Definition 5.4) and $E$ is anytime-valid admissible in the sense of Theorem 5.6;
- $\Delta \in \mathfrak{C}$ iff $\hat{C}$ is coverage-admissible per Definition 5.12, within the stated comparison class (Remark 5.15);
- $\Delta \in \mathfrak{D}$ iff $\sigma$ is CApp boundary-feasible per Definition 6.7.

At the component level, each criterion's admissibility (or feasibility, for $\mathfrak{D}$) is defined on the relevant procedure class; the four lifted classes $\mathfrak{B}, \mathfrak{A}, \mathfrak{C}, \mathfrak{D} \subseteq \Sigma$ above are their coordinatewise lifts via Definition 5.1.

*Remark* 5.3 (Lifting and active coordinates)*.* Every classical procedure ($e$-process, conformal set, defensive forecaster, etc.) lifts to a predictive system by setting the unused components to their default-inactive values. The criterion separation theorem below (Theorem 5.21) requires *more*: each witness $\Delta \in \Sigma$ must be active in both the membership and the failure coordinate, so that the failure under the second criterion is a substantive dominance failure within an active coordinate rather than categorical inapplicability. We verify activeness case by case in the proof of Theorem 5.21.

## 5.2 Anytime-valid admissibility

**Definition 5.4** (Anytime-valid constraint class)**.** Let $\mathcal{H}_0$ be a composite null. The *anytime-valid class* is

$$\mathcal{C}_{\mathrm{AV}} = \big\{(E_t)_{t\geq 1} : E_t \geq 0, \ \sup_{\mathbb{P}\in\mathcal{H}_0} \mathbb{E}_{\mathbb{P}}[E_\tau] \leq 1 \text{ for every stopping time } \tau\big\}.$$

Elements of $\mathcal{C}_{\mathrm{AV}}$ are called *e-processes*. By Ville's inequality (Ville, 1939), every $E \in \mathcal{C}_{\mathrm{AV}}$ provides anytime-valid type-I error control at level $\alpha$.

**Definition 5.5** (Admissibility within $\mathcal{C}_{\mathrm{AV}}$)**.** For $E, E' \in \mathcal{C}_{\mathrm{AV}}$, say $E'$ *dominates* $E$, written $E' \succeq_{\mathrm{AV}} E$, if $E'_t \geq E_t$ almost surely for every $t$ under every $\mathbb{P} \in \mathcal{H}_0$, with strict inequality at some time with positive probability under at least one $\mathbb{P} \in \mathcal{H}_0$. An element of $\mathcal{C}_{\mathrm{AV}}$ is *AV-admissible* if no element of $\mathcal{C}_{\mathrm{AV}}$ dominates it in this order.

**Theorem 5.6** (Ramdas et al. 2022)**.** *Within $\mathcal{C}_{\mathrm{AV}}$, a procedure is admissible if and only if it is a nonnegative martingale under every $\mathbb{P} \in \mathcal{H}_0$.*

*Remark* 5.7*.* Theorem 5.6 establishes admissibility relative to $\mathcal{C}_{\mathrm{AV}}$ and the criterion of type-I error control at every stopping time. This is distinct from Blackwell admissibility, which requires no domination under a loss $L(\theta, \delta)$ over all of $\Theta$. Figure 3 depicts the supermartingale cone, an admissible martingale path, and the stopped-process geometry behind Ville's inequality.

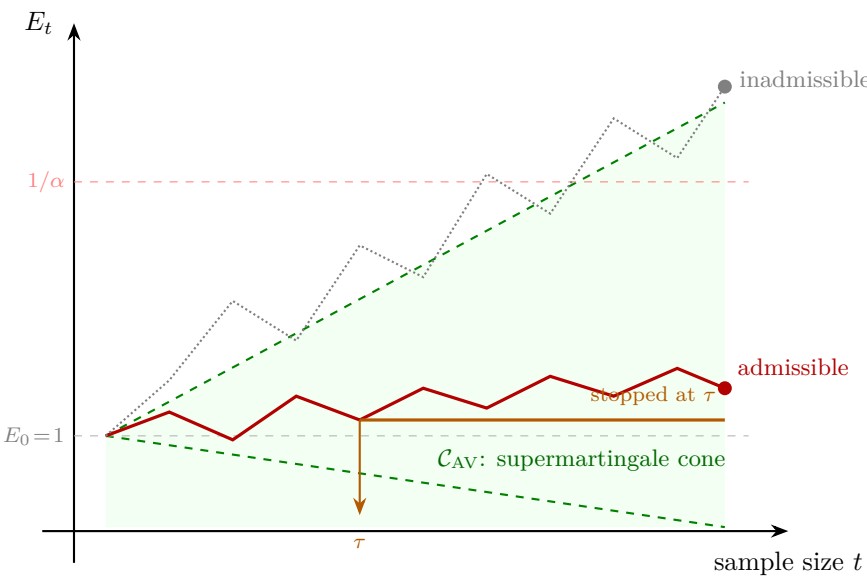

Figure 3: Supermartingale cone for anytime-valid inference. An e-process $E_t$ starts at $E_0 = 1$ and must remain a nonnegative supermartingale under every $\mathbb{P} \in \mathcal{H}_0$; the shaded cone depicts the feasible region $\mathcal{C}_{\mathrm{AV}}$. The solid line is an admissible e-process: a *nonnegative martingale* within the cone (Theorem 5.6). The dotted line is a process that grows systematically faster, violates the supermartingale condition under some $\mathbb{P} \in \mathcal{H}_0$, and is inadmissible. Stopping at any data-dependent time $\tau$ preserves type-I error control at level $\alpha$ via Ville's inequality: the stopped value $E_\tau \leq 1/\alpha$ with probability at least $1 - \alpha$ (downward arrow). All distinctions in the figure are encoded by line style so that the caption reads correctly in grayscale.

**Proposition 5.8** (Martingale as structural bridge)**.** *The martingale property relates to each admissibility criterion as follows:*

- *(i) Single-prior sequential Bayes $\Rightarrow$ martingale under the prior predictive: every posterior predictive sequence under a fixed prior $\Pi$ is a martingale under $\tilde{P} = \int P_\theta \, \mathrm{d}\Pi(\theta)$ (Proposition 4.3).*
- *(ii) AV-admissible $\Leftrightarrow$ nonnegative martingale within $\mathcal{C}_{\mathrm{AV}}$ (Theorem 5.6).*
- *(iii) Coverage validity does not require any martingale property: conformal prediction sets are constructed from rank statistics.*

*(iv) Self-consistency $\not\Rightarrow$ Blackwell admissible: the plug-in $\hat{p}_n^{\mathrm{pi}}$ is a martingale under its own predictive law $\hat{P}$ but $r(\hat{p}_n^{\mathrm{pi}}) \notin \partial_- \mathcal{R}$ (Theorem 4.4).*

*The martingale property is a complete characterization of AV-admissibility within $\mathcal{C}_{\mathrm{AV}}$, and is a structural feature of single-prior sequential Bayes coherence, but it is not a universal determinant of admissibility: it is neither necessary for fixed-sample Blackwell admissibility, nor needed for coverage validity or CApp boundary-feasibility.*

*Remark* 5.9 (Measure-relative martingale properties). The martingale property of a sequence $(M_n)$ is defined relative to a measure class, and the following three notions must be carefully distinguished:

(i) *Martingale under the prior predictive $\tilde{P} = \int P_\theta \, \mathrm{d}\Pi(\theta)$*: the Bayesian posterior predictive sequence satisfies this by the tower property (Proposition 4.3).
(ii) *Supermartingale under each $P \in \mathcal{H}_0$*: the defining property of e-processes in $\mathcal{C}_{\mathrm{AV}}$ (Definition 5.4), which guarantees anytime-valid error control via Ville's inequality.
(iii) *Self-predictive martingale under plug-in law $\hat{P}$*: the plug-in MLE $\hat{p}_n^{\mathrm{pi}}$ satisfies the martingale condition under $\hat{P}$ where $X_t \mid X_{1:t-1} \sim \mathrm{Bern}(\hat{p}_{t-1}^{\mathrm{pi}})$, yet is not admissible (Theorem 4.4).

Martingale coherence is not invariant across measures; admissibility depends on which measure class defines the risk functional.

*Remark* 5.10 (Certificate transport across filtrations). A related transport question arises *inside* the anytime-valid geometry itself. Choe & Ramdas (2026) study *e*-processes constructed in different filtrations and show that validity in a coarser filtration does not automatically carry over to a finer filtration; an *adjuster* is needed to lift the evidence process, and they prove that such adjusters are in a precise sense necessary. That obstruction is internal to $\mathcal{C}_{\mathrm{AV}}$: it concerns transporting a single anytime-valid certificate across information structures. Our separation theorem is orthogonal, concerning transport across *different criterion geometries*. Together the two results make the same methodological point from two directions: a certificate is tied both to an information structure and to a criterion, and neither dependence is automatic.

### 5.3 Marginal coverage admissibility

The third admissibility geometry concerns prediction sets rather than point predictions or sequential tests; its symmetry assumption is exchangeability (De Finetti, 1937; Hewitt & Savage, 1955). Two notions of coverage must be distinguished. *Marginal coverage* averages over both the calibration data and the new test point: it asks whether the prediction set contains the true response with probability at least $1 - \alpha$ under the joint exchangeable distribution. *Conditional coverage* conditions on the observed test covariate $X_{n+1} = x$ and demands coverage at every $x$ individually. Marginal coverage is achievable by conformal methods; conditional coverage, as the next theorem shows, is not achievable without degenerate prediction sets.

**Definition 5.11** (Marginal coverage). Given an exchangeable sequence $(X_1, \ldots, X_n, X_{n+1})$, a prediction set $\hat{C}_n(X_{n+1})$ satisfies *marginal coverage at level* $1 - \alpha$ if

$$\mathbb{P}\big(Y_{n+1} \in \hat{C}_n(X_{n+1})\big) \geq 1 - \alpha.$$

The coverage condition by itself is a feasibility constraint, not an admissibility relation: the trivial set $\hat{C}_n \equiv \mathcal{Y}$ always satisfies it. We promote it to a genuine dominance order by adding expected length to the comparison.

**Definition 5.12** (Coverage admissibility at level $1 - \alpha$). Fix a coverage level $1 - \alpha \in (0, 1)$. The *coverage-feasible class* is

$$\mathcal{F}_{\mathrm{Cov}}(\alpha) = \Big\{ \hat{C} = (\hat{C}_n) : \mathbb{P}\big(Y_{n+1} \in \hat{C}_n(X_{n+1})\big) \geq 1 - \alpha \text{ for all } n \geq 1 \Big\}.$$

For $\hat{C}, \hat{C}' \in \mathcal{F}_{\mathrm{Cov}}(\alpha)$, write $\hat{C}' \preceq_{\mathrm{Cov}} \hat{C}$ if

$$\mathbb{E}\,|\hat{C}_n'| \leq \mathbb{E}\,|\hat{C}_n| \qquad \text{for every } n \geq 1,$$

with strict inequality at some $n$. A prediction-set process $\hat{C} \in \mathcal{F}_{\mathrm{Cov}}(\alpha)$ is *coverage-admissible* (at level $1 - \alpha$) if no $\hat{C}' \in \mathcal{F}_{\mathrm{Cov}}(\alpha)$ satisfies $\hat{C}' \preceq_{\mathrm{Cov}} \hat{C}$.

*Remark* 5.13 (Why fixed-level dominance, not a 2D Pareto order). Coverage at level $1 - \alpha$ is the validity constraint imposed by the practitioner; once it is met, overcoverage carries no benefit, and a 2D Pareto comparison on $(c, \ell) = (\mathbb{P}(Y \in \hat{C}), \mathbb{E}|\hat{C}|)$ would prevent any shorter valid set from dominating a wider one (since the wider one typically has larger $c$). We therefore treat the coverage threshold as a feasibility requirement and measure efficiency *within* the feasible class by expected length alone. Under Definition 5.12, the trivial choice $\hat{C}_n \equiv \mathcal{Y}$ has coverage $1 \geq 1 - \alpha$ but maximal expected length, so it is strictly dominated by every shorter feasible set; this resolves the original categorical mismatch between marginal coverage as a one-sided inequality and admissibility as a dominance relation. An equivalent "capped-coverage" formulation, in which the coverage credit is $\min\{\mathbb{P}(Y \in \hat{C}), 1 - \alpha\}$, induces the same order on $\mathcal{F}_{\text{Cov}}(\alpha)$.

**Proposition 5.14** (Conformal sets are coverage-admissible within a score-threshold family). *Fix a nonconformity score $s : \mathcal{X} \times \mathcal{Y} \to \mathbb{R}$ and consider the comparison class $\mathcal{G}_s = \{\hat{C}^{(q)} : q \in \mathbb{R}\}$ of score-thresholded sets, with $\hat{C}_n^{(q)} = \{y : s(X_{n+1}, y) \leq q\}$. Restricting Definition 5.12 to comparisons within $\mathcal{G}_s$, the split-conformal threshold $q = \hat{q}_{1-\alpha}$ is coverage-admissible at level $1 - \alpha$.*

*Remark* 5.15 (Coverage admissibility is comparison-class relative). Coverage admissibility, like every efficiency notion, is relative to a specified comparison class $\mathcal{G}$ of prediction-set processes. The natural choice in conformal practice is the score-threshold family $\mathcal{G}_s$ used in Proposition 5.14. Throughout this paper, when we say a system $\Delta \in \Sigma$ lies in $\mathfrak{C}$, we mean its prediction-set coordinate is coverage-admissible within a stated comparison class (by default $\mathcal{G}_s$ for the canonical nonconformity score $s$ of the model); the separation witnesses below keep $\mathcal{G}_s$ constant across the systems being compared.

*Proof.* The family is nested: $q \leq q'$ implies $\hat{C}_n^{(q)} \subseteq \hat{C}_n^{(q')}$, so $q \mapsto \mathbb{E}|\hat{C}_n^{(q)}|$ is weakly increasing in $q$. By construction, $\hat{q}_{1-\alpha}$ is the smallest threshold in the score family for which $\mathbb{P}(s(X_{n+1}, Y_{n+1}) \leq \hat{q}_{1-\alpha}) \geq 1 - \alpha$ under exchangeability of the calibration and test data. Any threshold $q < \hat{q}_{1-\alpha}$ produces a shorter set but drops coverage below $1 - \alpha$, leaving $\mathcal{F}_{\text{Cov}}(\alpha)$. Any threshold $q > \hat{q}_{1-\alpha}$ remains in $\mathcal{F}_{\text{Cov}}(\alpha)$ but, by monotonicity of expected length, is weakly longer, so cannot satisfy $\hat{C}^{(q)} \preceq_{\text{Cov}} \hat{C}^{(\hat{q}_{1-\alpha})}$. Hence no element of the family dominates the conformal choice in the length order. $\square$

**Theorem 5.16** (Foygel Barber et al. 2021). *In the distribution-free setting, no nontrivial finite-length prediction set can guarantee exact conditional coverage*

$$\mathbb{P}(Y_{n+1} \in \hat{C}_n(X_{n+1}) \mid X_{n+1} = x) \geq 1 - \alpha$$

*uniformly over all distributions and at every non-atom $x$ of the covariate marginal. In the standard formulation, distribution-free conditional coverage forces prediction sets to be essentially uninformative—of infinite expected length at non-atoms of a continuous covariate distribution.*

The intuition is that distribution-free conditional coverage at every point $x$ would require the prediction set to accommodate the worst-case conditional distribution at each $x$ simultaneously; for continuous covariates this forces the set to grow without bound. The impossibility is specifically about *distribution-free conditional* coverage: it does not constrain marginal coverage, which conformal methods do attain, and it does not by itself adjudicate Blackwell optimality of a point predictor. It is one concrete face of the criterion separation that Theorem 5.21 formalizes; Figure 4 summarizes the coverage–width trade-off.

## 5.4 Criterion-relative admissibility

**Definition 5.17** (Criterion-relative admissibility). Write $\mathsf{Adm}_{\mathcal{C}}(\delta)$ to denote that $\delta$ is admissible relative to criterion $\mathcal{C}$: a comparison class of procedures together with the partial order induced by the relevant performance functional. The three criteria considered are:

(i) $\mathcal{C}_B$: Blackwell risk dominance: comparison class $\mathcal{D}$, ordering by coordinatewise risk dominance on $\mathcal{R}$, ambient geometry the convex risk set $\mathcal{R} \subset \mathbb{R}_+^k$.

(ii) $\mathcal{C}_{\text{AV}}$: anytime-valid admissibility: comparison class $\mathcal{C}_{\text{AV}}$ (Definition 5.4), ordering by expected stopped value, ambient geometry the cone of nonnegative supermartingales.

(iii) $\mathcal{C}_{\text{Cov}}$: marginal coverage at level $1 - \alpha$: comparison class a specified family $\mathcal{G}_s$ of prediction sets under exchangeability, ordering by expected length within the fixed-level coverage-feasible class $\mathcal{F}_{\text{Cov}}(\alpha) = \{\hat{C} : \mathbb{P}(Y_{n+1} \in \hat{C}) \geq 1 - \alpha\}$, ambient geometry the length frontier of $\mathcal{F}_{\text{Cov}}(\alpha) \cap \mathcal{G}_s$.

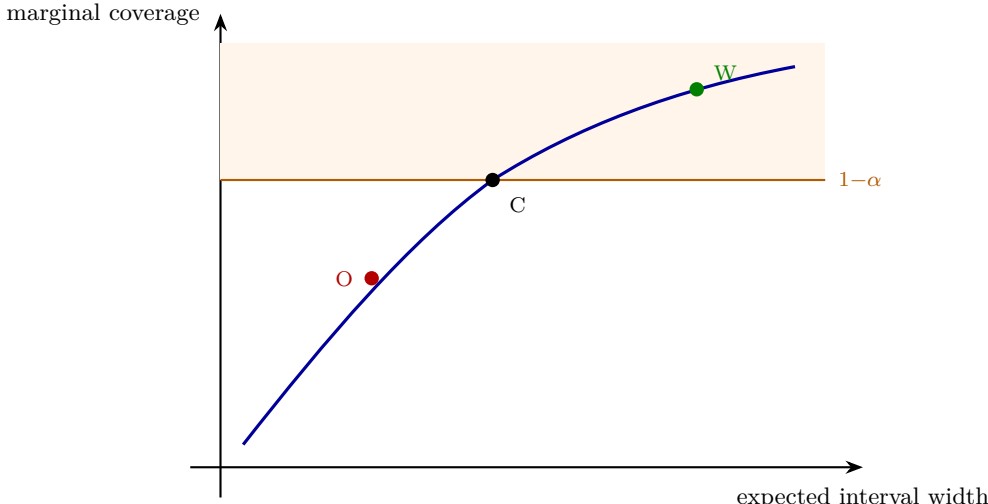

**Legend.** The shaded band above the horizontal line at $1-\alpha$ is the coverage-feasible class $\mathcal{F}_{\mathrm{Cov}}(\alpha)$ (Definition 5.12); the curve is the achievable coverage as a function of expected interval width. $\mathbf{C} = \hat{C}_n^{\mathrm{conf}}$: the split-conformal set at level $1 - \alpha$, on the frontier: a representative exchangeability-based set attaining the target coverage, expected-length efficient within its score-threshold family (Proposition 5.14). $\mathbf{O} = \hat{C}_n^{\mathrm{oracle}}$: a shorter oracle Bayes interval that undercovers and so falls outside $\mathcal{F}_{\mathrm{Cov}}(\alpha)$. $\mathbf{W} = \hat{C}_n^{\mathrm{wide}}$: a conservative feasible set that wastes width. Exact conditional coverage at every $x$ simultaneously is impossible (Theorem 5.16).

Figure 4: Coverage-feasible region for prediction sets. The horizontal line marks the validity threshold $1 - \alpha$; the shaded band above is the feasible class $\mathcal{F}_{\mathrm{Cov}}(\alpha)$. Among the three points, $\mathbf{C}$ lies on the frontier (conformal at the target coverage), $\mathbf{O}$ lies below it (oracle Bayes, infeasible), and $\mathbf{W}$ lies above it (conservative, feasible but wider than $\mathbf{C}$).

In this notation, $\mathfrak{B} = \{\delta : \mathsf{Adm}_{\mathcal{C}_B}(\delta)\}$, $\mathfrak{A} = \{\delta : \mathsf{Adm}_{\mathcal{C}_{\mathrm{AV}}}(\delta)\}$, $\mathfrak{C} = \{\delta : \mathsf{Adm}_{\mathcal{C}_{\mathrm{Cov}}}(\delta)\}$.

## 5.5 Constrained Bayes as a design principle

Definition 5.17 makes precise the sense in which each admissibility criterion operates on its own space of procedures and its own partial order. We now observe that every criterion in Definition 5.17 can be viewed as an instance of a single optimization template in which Bayesian risk is the objective and the validity requirement is a feasibility constraint.

**Definition 5.18** (Constrained Bayes schema). Each criterion $C \in \{B, \mathrm{AV}, \mathrm{Cov}, \mathrm{App}\}$ carries its own criterion-specific decision space $\mathcal{D}_C$ (the class of objects on which $C$ is defined: point predictors for Blackwell, $e$-processes for AV, prediction sets for Cov, online strategies for App), its own criterion-specific risk functional $\mathcal{R}_C(\theta, \cdot)$, and its own feasibility set $\mathcal{F}_C \subseteq \mathcal{D}_C$ encoding the validity requirement of $C$. Given a prior $\Pi$ on $\Theta$, the *constrained Bayes problem under criterion $C$* is

$$\min_{\delta \in \mathcal{D}_C} \int_{\Theta} \mathcal{R}_C(\theta, \delta) \, \mathrm{d}\Pi(\theta) \qquad \text{subject to} \qquad \delta \in \mathcal{F}_C. \tag{2}$$

A solution $\delta^*_{\mathcal{F}_C}$ is a *constrained Bayes rule for criterion $C$*. When $C = B$ and $\mathcal{F}_B = \mathcal{D}_B = \mathcal{D}$ is the unconstrained point-predictor space, the problem reduces to the unconstrained Bayes problem of Theorem 3.8.

Equivalently, lifting each component to $\Sigma$ (Definition 5.1), the schema is

$$\min_{\Delta \in \mathcal{F}_C \subseteq \Sigma} \int_{\Theta} \mathcal{R}_C(\theta, \Delta) \, \mathrm{d}\Pi(\theta),$$

where $\mathcal{R}_C$ depends only on the $C$-relevant coordinate of $\Delta$ and $\mathcal{F}_C$ requires activeness plus feasibility in that coordinate.

*Remark* 5.19 (The schema is not a single optimization problem). The four problems (2) are not subproblems of a single optimization on a common decision space: $\mathcal{D}_B$ contains point predictors, $\mathcal{D}_{\mathrm{AV}}$ contains $e$-processes, $\mathcal{D}_{\mathrm{Cov}}$ contains prediction sets, $\mathcal{D}_{\mathrm{App}}$ contains online strategies, and the risk functionals $\mathcal{R}_B, \mathcal{R}_{\mathrm{AV}}, \mathcal{R}_{\mathrm{Cov}}, \mathcal{R}_{\mathrm{App}}$ measure different things. The schema records that all four criteria share a common *template* (objective: Bayesian integrated risk; constraint: criterion-specific feasibility), but this is a recipe rather than a unifying theorem. The non-nesting in Theorems 5.21 and 6.12 arises precisely because the decision spaces, partial orders, and risk functionals differ across criteria.

*Remark* 5.20 (How the prior $\Pi$ enters the schema). Within the constrained Bayes schema two roles for the prior $\Pi$ must be carefully distinguished.

*(1) $\Pi$ as objective weight, $\mathcal{F}_C$ held fixed.* Holding the feasibility set $\mathcal{F}_C$ fixed, the prior $\Pi$ enters only the objective in (2); changing $\Pi$ does *not* change feasibility. Geometrically, $\Pi$ is the supporting-hyperplane normal of the restricted risk set $\mathcal{R}_{\mathcal{F}_C}$ (equivalently, a vector of shadow prices), and rotating it moves the constrained Bayes solution along the accessible frontier $\partial_- \mathcal{R}_{\mathcal{F}_C}$, exactly the role it plays on the unconstrained $\partial_- \mathcal{R}$.

*(2) $\Pi$ also entering $\mathcal{F}_C$.* If $\Pi$ also appears in the definition of $\mathcal{F}_C$ (for example, through a Bayesian credible-set construction that prescribes a set estimator built from the posterior under $\Pi$), then changing $\Pi$ changes both the objective and the feasible set. This is no longer the same constrained Bayes problem but a different one; the relationship between solutions across priors is then governed by joint sensitivity of the objective and the constraint, not by hyperplane rotation alone.

In the rest of the paper we work in regime (1) wherever possible: the feasibility constraints $\mathcal{F}_{\mathrm{AV}}, \mathcal{F}_{\mathrm{Cov}}, \mathcal{F}_{\mathrm{App}}$ are prior-free, and changing $\Pi$ moves the solution along $\partial_- \mathcal{R}_{\mathcal{F}_C}$ without altering $\mathcal{F}_C$.

The four admissibility geometries of this paper correspond to four choices of $(\mathcal{D}_C, \mathcal{F}_C, \mathcal{R}_C)$. Under Blackwell admissibility, $\mathcal{D}_B = \mathcal{D}$ (point predictors), $\mathcal{F}_B = \mathcal{D}$ (no constraint), $\mathcal{R}_B(\theta, \delta) = R(\theta, \delta)$; by Theorem 3.8 every solution lies on $\partial_- \mathcal{R}$. For anytime-valid inference, $\mathcal{D}_{\mathrm{AV}} = \mathcal{C}_{\mathrm{AV}}$ (the $e$-process cone), $\mathcal{F}_{\mathrm{AV}} = \mathcal{C}_{\mathrm{AV}}$ itself with the supermartingale constraint built in, and $\mathcal{R}_{\mathrm{AV}}$ is the expected stopped value at a stopping time (Theorem 5.6); admissibility reduces to the nonnegative martingale property. For marginal coverage, $\mathcal{D}_{\mathrm{Cov}}$ is the space of prediction-set processes, $\mathcal{F}_{\mathrm{Cov}}(\alpha) = \{\hat{C} : \mathbb{P}(Y_{n+1} \in \hat{C}_n) \geq 1 - \alpha\}$ (Definition 5.12), and $\mathcal{R}_{\mathrm{Cov}}$ is expected set length within the level-$(1-\alpha)$ feasible class. For CApp boundary-feasibility, $\mathcal{D}_{\mathrm{App}}$ is the space of online strategies, $\mathcal{F}_{\mathrm{App}}$ is the set of CApp-feasible strategies (Definition 6.7), and $\mathcal{R}_{\mathrm{App}}$ is the limsup of $\bar{R}_n - \partial_- \mathcal{R}$. The prior $\Pi$ enters each problem as the objective weight, not as a constraint.

When a constraint $\mathcal{F}_C \subsetneq \mathcal{D}_C$ is imposed, the optimization in (2) takes place over the restricted risk set $\mathcal{R}_{\mathcal{F}_C} = \{r_C(\delta) : \delta \in \mathcal{F}_C\} \subseteq \mathcal{R}_C$. The lower boundary of $\mathcal{R}_{\mathcal{F}_C}$ need not coincide with $\partial_- \mathcal{R}$: a constrained Bayes rule may fail to be Blackwell admissible because $\partial_- \mathcal{R}_{\mathcal{F}}$ may contain points in the interior of $\mathcal{R}$, and a Blackwell admissible rule may be infeasible because $r(\delta_\Pi)$ may lie outside $\mathcal{R}_{\mathcal{F}}$. Bayes remains the primal objective in every instance; the feasibility constraint determines which portion of the risk frontier is accessible, and thereby which priors can be realized as supporting hyperplanes of $\mathcal{R}_{\mathcal{F}}$.

This formulation makes the criterion-separation theorems below structurally inevitable. The four problems share the same objective form (minimize Bayesian risk), but the constraint sets $\mathcal{D}, \mathcal{C}_{\mathrm{AV}}, \mathcal{C}_{\mathrm{Cov}}, \mathcal{C}_{\mathrm{App}}$ are defined on different spaces of objects and induce different partial orders; none of the four orders subsumes any other. A single *component* cannot carry all four certificates at once without changing its semantic type: a point predictor is not an $e$-process, an $e$-process is not a prediction set, and so on. A modular predictive system $\Delta = (\delta, E, \hat{C}, \sigma) \in \Sigma$ can, however, combine components that satisfy several criteria componentwise (see Section 5.7 for explicit intersections); the admissibility claims then attach to different coordinates, and no one criterion subsumes the others. The non-nesting of admissible classes is therefore a consequence of incompatible feasibility sets attached to different coordinates, not a terminological artifact.

## 5.6 Separation theorem

**Intuition for Theorem 5.21.** Each of the three admissibility classes lives on a different coordinate of the predictive system $\Delta = (\delta, E, \hat{C}, \sigma)$: Blackwell admissibility on $\delta$, AV-admissibility on $E$, coverage admissibility on $\hat{C}$. The theorem says that within this single ambient space we can exhibit a predictive system that passes

one admissibility test on its coordinate but fails another admissibility test on a *different* coordinate, and that this failure can be made substantive (a dominance failure within the relevant coordinate) rather than categorical (the failure coordinate is simply absent). No one criterion subsumes another; joint optimality must be specified one coordinate at a time.

**Theorem 5.21** (Criterion separation on $\Sigma$). *Let $\mathfrak{B}$, $\mathfrak{A}$, $\mathfrak{C} \subseteq \Sigma$ denote the classes of predictive systems (Definition 5.1) that are Blackwell admissible, anytime-valid admissible, and coverage-admissible, respectively, in the sense of Definition 5.2. For each ordered pair $(\mathfrak{X}, \mathfrak{Y}) \subseteq \{\mathfrak{B}, \mathfrak{A}, \mathfrak{C}\}^2$ with $\mathfrak{X} \neq \mathfrak{Y}$, there exists a predictive system $\Delta \in \Sigma$ with components active in both the $\mathfrak{X}$- and $\mathfrak{Y}$-coordinates such that $\Delta \in \mathfrak{X}$ and $\Delta \notin \mathfrak{Y}$. The non-nestedness is therefore structural and substantive, not categorical.*

The table below previews the six witnesses, so the reader need not reconstruct them from the prose. Each is a single predictive system $\Delta \in \Sigma$; in each row the membership coordinate certifies the first class while a second, active coordinate fails the second class.

| Separation | Membership coordinate | Active failure coordinate |
|---|---|---|
| $\mathfrak{B} \not\subseteq \mathfrak{A}$ | Bayes predictor $\delta^B$ | strict supermartingale $E^\lambda$ |
| $\mathfrak{A} \not\subseteq \mathfrak{B}$ | LR martingale $E^{\mathrm{LR}}$ | constant-zero predictor $\delta \equiv 0$ |
| $\mathfrak{B} \not\subseteq \mathfrak{C}$ | Bayes predictor $\delta^B$ | over-wide set $\hat{C}^{(q')}$ |
| $\mathfrak{C} \not\subseteq \mathfrak{B}$ | conformal set $\hat{C}^{\mathrm{conf}}$ | plug-in predictor $\delta^{\mathrm{pi}}$ |
| $\mathfrak{A} \not\subseteq \mathfrak{C}$ | LR martingale $E^{\mathrm{LR}}$ | over-wide set $\hat{C}^{(q')}$ |
| $\mathfrak{C} \not\subseteq \mathfrak{A}$ | conformal set $\hat{C}^{\mathrm{conf}}$ | strict supermartingale $E^\lambda$ |

*Proof.* Fix a compact parameter set $\Theta = [\eta, 1 - \eta] \subset (0, 1)$ for some $\eta \in (0, \frac{1}{2})$ and the i.i.d. Bernoulli model $X_i \overset{\mathrm{iid}}{\sim} \mathrm{Bern}(\theta)$ with $\theta \in \Theta$, together with $\alpha \in (0, 1)$, $\theta_0 \in \Theta$. Let $\Pi_J$ denote the $\mathrm{Beta}(\frac{1}{2}, \frac{1}{2})$ (Jeffreys) prior restricted and renormalized to $\Theta$, so $\Pi_J$ has full topological support on $\Theta$. Define:

- $\delta_n^B = \mathbb{E}_{\Pi_J}[\theta \mid X_1, \ldots, X_n]$: the Bayes posterior predictive under the Jeffreys prior restricted and renormalized to $\Theta$ (the unrestricted closed form $(S_n + \frac{1}{2})/(n + 1)$ holds only on the full Bernoulli model $\Theta = (0, 1)$; the truncated prior gives a different but still $\mathcal{F}_n$-measurable posterior mean);
- $\delta_n^{\mathrm{pi}} = S_n/n$: plug-in MLE;
- $E_n^{\mathrm{LR}} = \prod_{t=1}^n (\delta_{t-1}^B/\theta_0)^{X_t}((1 - \delta_{t-1}^B)/(1 - \theta_0))^{1-X_t}$: likelihood-ratio e-process testing $H_0 : \theta = \theta_0$;
- $\hat{C}_n^{\mathrm{conf}}$: split-conformal prediction set with score $s(x, y) = |y - S_n/n|$ at level $1 - \alpha$;
- $\sigma^{\mathrm{act}}$: any fixed active strategy distinct from the default-inactive constant $a_\varnothing$, for instance the constant strategy $\sigma_n^{\mathrm{act}} \equiv a_1$ with $a_1 \neq a_\varnothing$. Defensive forecasting is not needed here and is not introduced until Section 6.

Each witness below is a predictive system $\Delta \in \Sigma$ whose components are active across all four coordinates; the failure verified in each case is a substantive dominance failure within the relevant coordinate.

*(i) $\mathfrak{B} \not\subseteq \mathfrak{A}$.* Take $\Delta'_B = (\delta^B, E^\lambda, \hat{C}^{\mathrm{conf}}, \sigma^{\mathrm{act}})$ with $E_n^\lambda := \lambda^n E_n^{\mathrm{LR}}$ for any fixed $\lambda \in (0, 1)$. Then $E^\lambda \geq 0$ and $\mathbb{E}[E_{n+1}^\lambda \mid \mathcal{F}_n] = \lambda^{n+1} E_n^{\mathrm{LR}} = \lambda E_n^\lambda < E_n^\lambda$ under every $P \in \mathcal{H}_0$, so $E^\lambda$ is a strict supermartingale: it lies in $\mathcal{C}_{\mathrm{AV}}$ but, not being a nonnegative martingale, is inadmissible within $\mathcal{C}_{\mathrm{AV}}$ by Theorem 5.6 (it is strictly dominated by $E^{\mathrm{LR}}$). Hence $E^\lambda \notin \mathfrak{A}$. The predictor $\delta^B$ is Blackwell admissible by Corollary 3.14 (compact $\Theta$ with the full-support prior $\Pi_J$ and continuous risk under log loss on $\Theta$), so $\Delta'_B \in \mathfrak{B}$. Both the predictor and the *e*-process coordinates are active.

$\mathfrak{A} \not\subseteq \mathfrak{B}$. Take $\Delta'_A = (\delta', E^{\mathrm{LR}}, \hat{C}^{\mathrm{conf}}, \sigma^{\mathrm{act}})$ where $\delta' \equiv 0$ is the constant zero predictor. The predictor is feasible but Blackwell-inadmissible (its log-loss risk is $+\infty$ at every $\theta > 0$, dominated by $\delta^B$); hence $\Delta'_A \notin \mathfrak{B}$. The e-process $E^{\mathrm{LR}}$ is a nonnegative martingale under $H_0$, hence AV-admissible by Theorem 5.6, giving $\Delta'_A \in \mathfrak{A}$. Both predictor and *e*-process components are active (the constant zero predictor is a definite choice, not the default-inactive placeholder constant $a_\varnothing$ used to lift).

*(ii) $\mathfrak{B} \not\subseteq \mathfrak{C}$.* Modify $\Delta'_B$ by replacing $\hat{C}^{\mathrm{conf}} = \hat{C}^{(\hat{q}_{1-\alpha})}$ with $\hat{C}^{(q')}$ for a strictly larger threshold $q' > \hat{q}_{1-\alpha}$ in the same score-threshold family $\mathcal{G}_s$, chosen so that the enlargement is active in the family: $\mathbb{E}|\hat{C}_n^{(q')}| > \mathbb{E}|\hat{C}_n^{(\hat{q}_{1-\alpha})}|$

for at least one $n$. Then $\hat{C}^{(q')} \in \mathcal{F}_{\mathrm{Cov}}(\alpha)$ (coverage at level $1 - \alpha$ is preserved by enlarging the set) and $\hat{C}^{(\hat{q}_{1-\alpha})} \preceq_{\mathrm{Cov}} \hat{C}^{(q')}$ strictly within $\mathcal{G}_s$, so $\hat{C}^{(q')} \notin \mathfrak{C}$ by Definition 5.12. Both the predictor and the set coordinate are active within their respective comparison classes. The predictor remains $\delta^B$, so the system is in $\mathfrak{B}$.

$\mathfrak{C} \nsubseteq \mathfrak{B}$. Retain $\hat{C}^{\mathrm{conf}}$ (coverage-admissible by Proposition 5.14) and use the plug-in predictor $\delta^{\mathrm{pi}}$, which is Blackwell-inadmissible under log loss (Theorem 4.4). The system $(\delta^{\mathrm{pi}}, E^{\mathrm{LR}}, \hat{C}^{\mathrm{conf}}, \sigma^{\mathrm{act}})$ is therefore in $\mathfrak{C} \setminus \mathfrak{B}$.

*(iii)* $\mathfrak{A} \nsubseteq \mathfrak{C}$. Take $\Delta'_A$ from (i) and additionally replace its prediction-set component by the strictly wider $\hat{C}^{(q')}$ from (ii). Then $\Delta'_A \in \mathfrak{A}$ but its prediction-set component is not coverage-admissible within $\mathcal{G}_s$, so $\Delta'_A \notin \mathfrak{C}$.

$\mathfrak{C} \nsubseteq \mathfrak{A}$. Modify $(\delta^{\mathrm{pi}}, E^{\mathrm{LR}}, \hat{C}^{\mathrm{conf}}, \sigma^{\mathrm{act}})$ by replacing $E^{\mathrm{LR}}$ with the strict-supermartingale process $E_n^\lambda = \lambda^n E_n^{\mathrm{LR}}$ of (i). The prediction-set component remains coverage-admissible; the e-process is no longer AV-admissible. The system is in $\mathfrak{C} \setminus \mathfrak{A}$. $\qquad\square$

**Interpretation.** The theorem is not merely comparing procedures of different types and observing that they are different. Rather, it establishes that none of the three partial orders $\leq_{\mathcal{C}_B}, \leq_{\mathcal{C}_{\mathrm{AV}}}, \leq_{\mathcal{C}_{\mathrm{Cov}}}$ subsumes the others within $\Sigma$: no one criterion ranks all procedures in a way that recovers boundary optimality under the other two. Trivially, one can always concoct artificial product or lexicographic orders on $\Sigma$ that contain the three partial orders as restrictions; the substantive content of the theorem is that no *canonical, criterion-neutral* refinement is supplied by the three theories themselves. Consequently, any meta-criterion that ranks all procedures across the three frameworks must adjoin an extra normative choice not present in their definitions. The non-nesting arises from the geometry of the constraint sets, not from terminology, and it persists for any statistical model that admits all three inferential tasks.

*Remark* 5.22 (No filtration trick). The non-nesting in Theorem 5.21 is not a matter of choosing a different stopping time or filtration. Blackwell dominance, AV-admissibility, and coverage admissibility act on different coordinates of $\Sigma$ and are certified by different mathematical objects.

### 5.7 Compatibility and relative size of the admissible classes

Non-nesting of $\mathfrak{B}, \mathfrak{A}, \mathfrak{C}$ within $\Sigma$ does *not* imply that these classes are disjoint; the criterion-separation theorem only forbids subsumption. In practice the four classes typically have nonempty intersections, and assembling a procedure that lies in several at once is a matter of design rather than an impossibility.

**Intersections are routinely nonempty.** Three concrete intersections illustrate the point.

(i) A predictive system whose *e*-process is the likelihood ratio under the Bayes posterior predictive and whose prediction set is the split-conformal set built from the same predictor lies in $\mathfrak{A} \cap \mathfrak{C}$: it controls type-I error at every stopping time and is expected-length efficient within its score-threshold family in $\mathcal{F}_{\mathrm{Cov}}(\alpha)$ (Proposition 5.14).

(ii) A Bayes predictor under a full-support prior, wrapped by a split-conformal procedure, lies in $\mathfrak{B} \cap \mathfrak{C}$: admissibility of the predictor coordinate is supplied by Corollary 3.12; admissibility of the set coordinate is supplied by Proposition 5.14.

(iii) On a compact Bernoulli submodel $\Theta = [\eta, 1 - \eta] \subset (0,1)$, the Bayes posterior predictive under a smooth full-support prior such as the Jeffreys prior restricted and renormalized to $\Theta$ lies in $\mathfrak{B}$ by Corollary 3.14. Standard log-loss redundancy bounds for regular one-dimensional Bernoulli mixtures (Cesa-Bianchi & Lugosi, 2006, §9.7) give Cesàro convergence of its time-averaged risk to $H(\theta)$, hence CApp boundary-feasibility, so the lifted system is also in $\mathfrak{D}$. The KT formula $\hat{p}_n^{\mathrm{KT}} = (S_n + \frac{1}{2})/(n + 1)$ provides the simplest explicit version of the same phenomenon on the full Bernoulli model. Thus $\mathfrak{B} \cap \mathfrak{D} \neq \varnothing$; constructive and Cesàro admissibility *can* coincide.

What the separation theorem rules out is not coexistence but *universal subsumption*: no one of the four criteria provides a certificate that automatically implies the other three.

**Relative size: a geometric remark.** Although we do not state a formal cardinality or measure-theoretic result, the four classes have visibly different geometric shapes within $\Sigma$, and a brief orientation helps.

The Blackwell class $\mathfrak{B}$ is the lower boundary of a convex risk set: it is a *thin* object (a $(k-1)$-dimensional boundary in $\mathbb{R}^k$ for finite $|\Theta| = k$), and every interior risk vector is dominated. Anytime-valid admissibility within $\mathcal{C}_{\text{AV}}$ is the cone of nonnegative martingales under $\mathcal{H}_0$; admissible elements form a strictly smaller subcone than $\mathcal{C}_{\text{AV}}$ itself (strict supermartingales are feasible but inadmissible, as in the witness $E_n^\lambda = \lambda^n E_n^{\text{LR}}$ used in the proof of Theorem 5.21). Marginal coverage as a feasibility condition ($\hat{C} \in \mathcal{F}_{\text{Cov}}(\alpha)$ with $\mathbb{P}(Y \in \hat{C}) \geq 1 - \alpha$) describes a substantially larger object than the Blackwell boundary: it is an entire half-space-like region of $\Sigma$, since any sufficiently large prediction set is feasible. Once efficiency within $\mathcal{F}_{\text{Cov}}(\alpha)$ is measured by expected length (Definition 5.12), the analogy with Blackwell admissibility tightens: both become thin boundary objects, but in incomparable geometries (loss risk versus the fixed-level coverage frontier ordered by expected length).

The CApp class $\mathfrak{D}$ is the set of online strategies whose time-averaged risk reaches $\partial_- \mathcal{R}$ in the limit; this is neither a boundary of a static risk set nor a feasibility region, but an asymptotic statement. Its geometry is best thought of as the set of trajectories converging to a target rather than a static subset of a Euclidean space.

In short: coverage *feasibility* is large compared with the Blackwell boundary, but coverage *admissibility* after expected-length refinement is a comparably thin frontier; AV-admissibility and CApp-feasibility are thin within their respective ambient geometries. The reason no master ordering exists is that the four boundary geometries live in categorically different ambient spaces.

## 6 Constructive and Choice-Based Admissibility

The connection between Blackwell approachability (Blackwell, 1956) and no-regret learning (Hannan, 1957; Abernethy et al., 2011) invites a question: does steering the time-averaged risk to the lower boundary suffice for admissibility? This section distinguishes two routes to $\partial_- \mathcal{R}$ and shows that the distinction generates a fourth admissibility geometry.

### 6.1 Two paths to the boundary

**Definition 6.1** (Constructive admissibility)**.** A procedure $\delta$ is *constructively admissible* if there exists a prior $\Pi_n$ at every sample size $n$ such that $\delta(X^n) = \delta_{\Pi_n}(X^n)$, i.e., $\delta$ is Bayes with respect to an explicitly specified, sample-size-dependent prior at each round.

This is a pointwise (per-round) boundary condition: each action is itself boundary-certified by the supporting hyperplane of its prior (Theorem 3.8).

**Definition 6.2** (Cesàro admissibility)**.** A procedure $\delta$ is *Cesàro admissible* if the time-averaged risk $\bar{R}_n(\theta, \delta) = n^{-1} \sum_{t=1}^n R(\theta, \delta_t)$ converges to the oracle envelope $\partial_- \mathcal{R}(\theta)$ as $n \to \infty$ (the per-$\theta$ envelope sense made precise in Definition 6.7), without requiring that each individual action $\delta_t$ be Bayes optimal at time $t$.

This is a Cesàro boundary condition: only the time-average risk approaches the envelope $\partial_- \mathcal{R}(\theta)$, so individual rounds may lie in the interior of the risk set.

Constructive admissibility demands a per-round witness (the prior $\Pi_n$), while Cesàro admissibility requires only that the long-run average reaches the boundary. The Blackwell approachability theorem guarantees the existence of Cesàro-admissible strategies whenever the target set is approachable; it does not, in general, produce constructively admissible ones.

### 6.2 Approachability revisited: the missing martingale layer

Abernethy et al. (2011) show that Blackwell approachability and no-regret learning are equivalent. If $S = \mathcal{R}$ is the risk set, approachability of $\partial_- \mathcal{R}$ guarantees $\bar{R}_n \to \partial_- \mathcal{R}$, but the equivalence operates in the Cesàro regime: it does not require that each per-round action $\delta_t$ be individually Bayes optimal. The martingale layer (Section 4) provides the missing intertemporal constraint: the prior sequence $(\Pi_t)$ must update coherently via Bayes' rule, not merely converge in Cesàro average. Approachability ensures arrival at the boundary; the martingale property ensures the journey is coherent.

The asymptotic calibration theorem of Foster & Vohra (1998) illustrates the gap: calibration error vanishes in the Cesàro sense, but the per-round action rule is selected by a fixed-point argument (Kakutani's theorem, or the continuous-selection refinements of Hart & Mas-Colell (2001)), not by posterior updating. Foster–Vohra calibration therefore sits naturally in the CApp geometry (Definition 6.7): the boundary is reached by choice-based dynamics, with no per-round Bayes witness required.

### 6.3 Cesàro does not imply pointwise

*Remark* 6.3 (Choice-based strategies illustrate Cesàro guarantees). Choice-based calibration procedures such as the Foster–Vohra algorithm (Foster & Vohra, 1998) and the defensive forecaster (Vovk et al., 2005b) illustrate Cesàro-style guarantees under a calibration-error objective: their predictions $\hat{p}_t$ are determined by a fixed-point argument (Kakutani's theorem or the continuous-selection refinements of Hart & Mas-Colell (2001)), not by posterior updating from a prior, and the empirical calibration error $|\bar{p}_n - \bar{X}_n|$ vanishes almost surely. We do *not* use these procedures as log-loss CApp-feasibility witnesses: as Remark 6.10 explains, calibration in the Cesàro sense does not imply convergence of the time-averaged log loss to the entropy floor $H(\theta)$, because log loss is strictly convex and Jensen's inequality breaks the implication. The log-loss CApp witness we rely on in this paper is the Krichevsky–Trofimov forecaster (Lemma 6.9).

The qualitative gap between Cesàro and pointwise admissibility is the sequential analogue of the gap between minimax and Bayes optimality: the former guarantees a limiting performance level, the latter provides a per-instance optimality certificate. The fixed-point argument is the *construction principle* (how the per-round action is selected); Cesàro convergence is the *mode of optimality* (the sense in which the resulting strategy reaches the boundary). The two should not be conflated.

### 6.4 Martingale characterization of constructive admissibility

**Proposition 6.4** (Single-prior martingale characterization). *Let $\delta = (\delta_n)$ be Bayes under a single prior $\Pi$ at every $n$, i.e. $\delta_n = \delta_\Pi$. Then the posterior predictive sequence $(m_n) = (\mathbb{E}_\Pi[\theta \mid \mathcal{F}_n])$ is a martingale under the prior predictive law $\tilde{P} = \int P_\theta \, d\Pi(\theta)$.*

*Proof.* Immediate from Proposition 4.3 applied to $\tilde{P}$. $\qquad\square$

**Proposition 6.5** (Sample-size-dependent martingale, with Kolmogorov compatibility). *Let $\delta = (\delta_n)$ satisfy $\delta_n = \delta_{\Pi_n}$ for a sequence of priors $\{\Pi_n\}_{n\geq 1}$. Define the prior predictive laws $\tilde{P}_n = \int P_\theta \, d\Pi_n(\theta)$ on $(\mathcal{X}^n, \mathcal{F}_n)$. If the family $\{\tilde{P}_n\}$ is projectively compatible,*

$$\tilde{P}_{n+1}\big|_{\mathcal{F}_n} = \tilde{P}_n \qquad \text{for all } n \geq 1,$$

*then by Kolmogorov's extension theorem there exists a single law $\tilde{P}_\infty$ on $(\mathcal{X}^\infty, \mathcal{F}_\infty)$ with $\tilde{P}_\infty|_{\mathcal{F}_n} = \tilde{P}_n$, and the posterior predictive sequence is a martingale under $\tilde{P}_\infty$.*

*Proof.* Projective compatibility is Kolmogorov's consistency condition. The extension $\tilde{P}_\infty$ exists by Kolmogorov's theorem on the Polish sample space $\mathcal{X}^\infty$ (Definition 2.1). The martingale property of $(m_n)$ under $\tilde{P}_\infty$ follows from the tower property: $\mathbb{E}_{\tilde{P}_\infty}[m_{n+1} \mid \mathcal{F}_n] = \mathbb{E}_{\tilde{P}_{n+1}}[\theta \mid \mathcal{F}_n] = m_n$, where the first equality uses $\tilde{P}_\infty|_{\mathcal{F}_{n+1}} = \tilde{P}_{n+1}$ and the second uses the definition of $m_n$. $\qquad\square$

*Remark* 6.6 (Single prior vs. varying priors). Proposition 6.4 is what the rest of this paper actually uses, and what Proposition 4.3 directly delivers. Proposition 6.5 addresses the subtler case in which the prior $\Pi_n$ is allowed to depend on the sample size: being Bayes at each $n$ under a possibly different prior does *not* imply the existence of a single prior predictive law under which the sequence is a martingale. The Kolmogorov-compatibility hypothesis is exactly the family-of-priors analogue of the consistency condition required for the extension to exist.

The single-prior version makes precise the sense in which the martingale layer separates constructive from Cesàro admissibility: the former requires the journey to be a martingale under a single law, while the latter only requires the destination.

### 6.5 A fourth geometry: choice-based approachability admissibility

**Definition 6.7** (CApp boundary-feasibility). Fix a loss function $L$ and let

$$\bar{R}_n(\theta, \sigma) \;=\; n^{-1} \sum_{t=1}^{n} L\big(\theta, \sigma_t(X_{1:t-1})\big)$$

denote the time-averaged risk of an online strategy $\sigma$. The strategy $\sigma$ is *CApp boundary-feasible* (or simply *CApp-feasible*) if

$$\mathrm{dist}\big(\bar{R}_n(\theta, \sigma), \partial_-\mathcal{R}(\theta)\big) \;\longrightarrow\; 0 \qquad \text{in } P_\theta\text{-probability as } n \to \infty, \text{ for every } \theta \in \Theta,$$

where $\partial_-\mathcal{R}(\theta)$ denotes the *oracle per-round envelope* at $\theta$, $\partial_-\mathcal{R}(\theta) = \mathrm{env}(\theta) := \inf_{a \in \mathcal{A}} L(\theta, a)$, which equals the binary entropy $H(\theta)$ under Bernoulli log loss. We write $\mathfrak{D}$ for the class of predictive systems whose strategy coordinate is CApp boundary-feasible. CApp boundary-feasibility is a coordinatewise asymptotic statement, one $\theta$ at a time; it is not the assertion that some finite-sample risk *vector* lies on the Pareto frontier $\partial_-\mathcal{R}$ of Section 3, and the two should not be conflated (cf. the envelope/boundary distinction in Example 2.4).

*Remark* 6.8 (Why feasibility, not full admissibility). The previous statement of this definition required only "$\bar{R}_n(\theta, \sigma) \to \partial_-\mathcal{R}$" without specifying which loss or which mode of convergence. The revised version pins down *the loss $L$*, which is part of the data of the decision problem, and *the mode of convergence*, in $P_\theta$-probability, which accommodates stochastic strategies. We deliberately stop at *feasibility* (boundary-reachability) rather than imposing a further dominance order on CApp-feasible strategies: proving asymptotic non-domination for KT-type strategies would require minimax/redundancy lower bounds beyond the scope of this paper, and our use of $\mathfrak{D}$ in Theorems 5.21–6.12 only needs feasibility, since the separation witnesses establish $\mathfrak{X} \not\subseteq \mathfrak{Y}$ and $\mathfrak{Y} \not\subseteq \mathfrak{X}$ by exhibiting concrete predictive systems in one class but not the other. Calibration in the empirical sense ($|\bar{p}_n - \bar{X}_n| \to 0$) is a property that implies CApp boundary-feasibility under squared loss but does *not* imply it under log loss; the log-loss CApp witness we use is the KT forecaster (Lemma 6.9, Remark 6.10).

For *calibration-error* objectives, the defensive forecaster (Vovk et al., 2005b) and the Foster–Vohra calibration algorithm (Foster & Vohra, 1998) are paradigmatic CApp examples, achieving calibration by choice-based fixed-point arguments; the adaptive strategies of Hart & Mas-Colell (2001) extend the family. Under *log loss*, we do not rely on these procedures (see Remark 6.10); the log-loss CApp witness we rely on is the KT forecaster, which is itself the Bayes posterior predictive under the Jeffreys prior. Before stating the extended separation we establish that the Krichevsky–Trofimov (KT) forecaster reaches the lower boundary $\partial_-\mathcal{R}$ in time-average under log loss in the i.i.d. Bernoulli model. We then state separately what is and is not known about defensive forecasting in the same setting; the KT case alone is sufficient for the separation arguments below.

**Lemma 6.9** (KT is CApp-feasible under log loss). *In the Bernoulli i.i.d. model under log loss, the Krichevsky–Trofimov forecaster (Krichevsky & Trofimov, 1981) $\hat{p}_n^{\mathrm{KT}} = (S_n + \frac{1}{2})/(n+1)$ (equivalently, the Bayes posterior predictive under the Jeffreys prior $\mathrm{Beta}(\frac{1}{2}, \frac{1}{2})$) satisfies $\bar{R}_n(\theta, \hat{p}^{\mathrm{KT}}) \to H(\theta)$ in $P_\theta$-probability for every $\theta \in (0,1)$, where $H(\theta) = -\theta \log \theta - (1-\theta)\log(1-\theta)$ is the binary entropy. Since $\partial_-\mathcal{R}(\theta) = H(\theta)$ under log loss, the KT forecaster is CApp-feasible.*

*Proof sketch.* The KT forecaster has cumulative log-loss regret of order $O(\log n)$ relative to the best constant predictor in hindsight (Cesa-Bianchi & Lugosi, 2006, §9.7). By the law of large numbers, the empirical best constant in hindsight converges $P_\theta$-a.s. to $\theta$, and its average log loss converges to $H(\theta)$. The cumulative martingale-difference term between the realized log loss at step $t$ and the conditional expected log loss given $\mathcal{F}_{t-1}$ is $o_p(n)$, hence its Cesàro average is $o_p(1)$. Combining these three pieces, the time-averaged conditional log-loss risk $\bar{R}_n(\theta, \hat{p}^{\mathrm{KT}})$ converges in $P_\theta$-probability to $H(\theta)$. □

*Remark* 6.10 (Defensive forecasting: calibration vs. log-loss boundary). The defensive forecaster of Vovk et al. (2005b) achieves asymptotic empirical calibration ($|\bar{p}_n - \bar{X}_n| \to 0$); this is a property of the time-average of predictions, not of the time-average of log losses. Because log loss is strictly convex, $n^{-1} \sum_t L(\theta, p_t) \neq L(\theta, n^{-1} \sum_t p_t)$ in general, so calibration alone does not imply convergence of the Cesàro-average log loss

to $H(\theta)$. A calibrated forecaster can be unsharp and incur excess log loss. Establishing log-loss CApp-feasibility for defensive forecasting therefore requires an additional regret guarantee under log loss (which is not the calibration statement used in Vovk et al. (2005b)). We do *not* rely on a log-loss boundary claim for defensive forecasting in this paper; the extended separation below uses the KT forecaster (Lemma 6.9) as the worked CApp-feasible witness, and Foster–Vohra calibration as a separate choice-based example under a calibration-error objective rather than log loss.

*Remark* 6.11 ($\mathfrak{B} \cap \mathfrak{D} \neq \emptyset$ on a compact Bernoulli submodel). On the compact Bernoulli submodel $\Theta = [\eta, 1-\eta]$, the Bayes posterior predictive under a smooth full-support prior (such as the Jeffreys prior restricted and renormalized to $\Theta$) is Blackwell admissible by Corollary 3.14. Standard log-loss redundancy bounds for regular one-dimensional Bernoulli mixtures (Cesa-Bianchi & Lugosi, 2006, §9.7) give Cesàro convergence of its time-averaged risk to $H(\theta)$, hence CApp boundary-feasibility (cf. Lemma 6.9). The KT forecaster $\hat{p}_n^{\mathrm{KT}} = (S_n + \frac{1}{2})/(n+1)$ is the simplest explicit version of the same phenomenon on the full Bernoulli model. Hence $\mathfrak{B} \cap \mathfrak{D} \neq \emptyset$, which is consistent with the separation theorem below: the non-nesting is a statement about *neither* containing the other, not about disjointness. The separations in Theorem 6.12 require either a misspecified prior (case (i) below) or a CApp-feasible strategy outside the Bayes class (case (ii)).

**Intuition for Theorem 6.12.** Adding the strategy coordinate $\sigma$ and the CApp class $\mathfrak{D}$ to the picture extends the separation in two directions. The Bayes-versus-CApp failure direction is illustrated by a misspecified-prior Bayes rule that stays admissible at a point to which the prior commits, yet whose time-averaged risk under the *true* parameter never reaches the entropy floor. The CApp-versus-Bayes failure direction is illustrated by a predictive system that pairs the KT online strategy (CApp boundary-feasible by Lemma 6.9) with the plug-in MLE predictor (Blackwell-inadmissible under log loss by Theorem 4.4); the failure is therefore a substantive dominance failure in the predictor coordinate, not a categorical mismatch. Cesàro boundary-reachability and per-round Bayes optimality are distinct certificates, and neither encompasses the other.

**Theorem 6.12** (Extended separation on $\Sigma$). *Let $\mathfrak{B}$, $\mathfrak{A}$, $\mathfrak{C}$, $\mathfrak{D}$ denote the classes of predictive systems $\Delta \in \Sigma$ (Definition 5.1) that satisfy the corresponding criterion property in Definition 5.2. The four classes are pairwise non-nested: for each ordered pair $(\mathfrak{X}, \mathfrak{Y})$ with $\mathfrak{X} \neq \mathfrak{Y}$, there exists a predictive system $\Delta \in \Sigma$ with components active in both the $\mathfrak{X}$- and $\mathfrak{Y}$-coordinates such that $\Delta \in \mathfrak{X}$ and $\Delta \notin \mathfrak{Y}$.*

*Proof.* The pairwise non-nesting of $\mathfrak{B}$, $\mathfrak{A}$, $\mathfrak{C}$ within $\Sigma$ is established in Theorem 5.21. It remains to verify the six separations involving $\mathfrak{D}$.

*(i) $\mathfrak{B} \nsubseteq \mathfrak{D}$: misspecified-prior witness.* Work on the same compact Bernoulli model $\Theta = [\eta, 1 - \eta]$ used in Theorem 5.21, chosen so that $0.3, 0.7 \in \Theta$, and let $\Pi = \delta_{0.3}$ be the point mass at $\theta = 0.3$. The Bayes rule under $\Pi$ at every $n$ is the constant predictor $\delta_\Pi \equiv 0.3$ (the posterior remains $\delta_{0.3}$ for all data; the Bayes act minimizing $R(0.3, a)$ over $a \in [0, 1]$ under log loss is $a = 0.3$). This predictor is Blackwell admissible by a direct argument (rather than via Corollary 3.14, which requires full topological support). Indeed, the lower-boundary value at $\theta = 0.3$ is $H(0.3)$, attained by the constant action $a = 0.3$. Any rule $\delta'$ satisfying $R(0.3, \delta') \leq R(0.3, \delta_\Pi) = H(0.3)$ must, by strict properness of log loss (Gneiting & Raftery, 2007), predict $a = 0.3$ $P_{0.3}$-a.s. Since $P_\theta$ and $P_{\theta'}$ are mutually absolutely continuous on $\{0, 1\}^n$ for every $\theta, \theta' \in (0, 1)$ (Bernoulli product measures with parameters in $(0, 1)$ have a common support), equality $P_{0.3}$-a.s. implies equality $P_\theta$-a.s. for every $\theta \in \Theta$. Hence $\delta'$ agrees with $\delta_\Pi$ on a set of $P_\theta$-measure one for each $\theta$ and cannot strictly improve anywhere on $\Theta$. No rule dominates $\delta_\Pi$, so $\delta_\Pi$ is Blackwell admissible.

Compute the per-round risk at the misspecified parameter $\theta^* = 0.7$:

$$R(0.7, \delta_\Pi) = -0.7 \log(0.3) - 0.3 \log(0.7)$$
$$\approx 0.8428 + 0.1070 = 0.9498 \text{ nats.}$$

The Cesàro average $\bar{R}_n(0.7, \delta_\Pi) \equiv 0.9498$ for all $n$ (the predictor never updates). The lower boundary at $\theta = 0.7$ is $H(0.7) = -0.7 \log(0.7) - 0.3 \log(0.3) \approx 0.6109$. Hence $\bar{R}_n(0.7, \delta_\Pi) - H(0.7) \approx 0.339$ nats per round, a gap that never closes, so $\mathrm{dist}(\bar{R}_n(0.7, \delta_\Pi), \partial_- \mathcal{R}) \nrightarrow 0$ and $\delta_\Pi \notin \mathfrak{D}$. The associated predictive system $\Delta = (\delta_\Pi, E^{\mathrm{LR}}, \hat{C}^{\mathrm{conf}}, \delta_\Pi)$ has active predictor, $e$-process, prediction-set, and strategy components. It satisfies $\Delta \in \mathfrak{B}$ (the predictor is Blackwell admissible) yet $\Delta \notin \mathfrak{D}$ (the strategy is not CApp-feasible). This

furnishes a substantive (non-categorical) witness of $\mathfrak{B} \not\subseteq \mathfrak{D}$, replacing the earlier "Bayes is not defined by approachability" sketch. The witness depends essentially on the dogmatic point mass, which no amount of data corrects: a full-support prior would concentrate at the truth and, by Remark 6.11, land in $\mathfrak{B} \cap \mathfrak{D}$. (For the misspecified-model analogue, where posteriors concentrate on the Kullback–Leibler projection, see Kleijn & van der Vaart 2012.)

*(ii) $\mathfrak{D} \not\subseteq \mathfrak{B}$.* Consider the predictive system

$$\Delta \;=\; \big(\delta^{\mathrm{pi}}, E^{\mathrm{LR}}, \hat{C}^{\mathrm{conf}}, \sigma^{\mathrm{KT}}\big),$$

where $\sigma^{\mathrm{KT}}$ is the KT online strategy $\sigma_n^{\mathrm{KT}} = (S_{n-1} + \frac{1}{2})/n$ (the Krichevsky–Trofimov forecaster used in Lemma 6.9) and $\delta_n^{\mathrm{pi}} = S_n/n$ is the plug-in MLE predictor. The strategy coordinate is CApp-feasible under log loss by Lemma 6.9, so $\Delta \in \mathfrak{D}$ via Definition 5.2. The predictor coordinate $\delta^{\mathrm{pi}}$ is Blackwell-inadmissible under log loss by Theorem 4.4, so $\delta^{\mathrm{pi}} \notin \mathfrak{B}$ and hence $\Delta \notin \mathfrak{B}$. Both the predictor and the strategy coordinates are active: $\delta^{\mathrm{pi}}$ differs from any constant predictor on a positive-measure event, and $\sigma^{\mathrm{KT}}$ differs from any constant strategy at every $n \geq 1$. The failure of $\mathfrak{B}$ is therefore a *substantive* dominance failure in the predictor coordinate, not a categorical mismatch, exactly as the $\Sigma$-space formulation requires.

(An interpretive aside, not used in the separation proof: the Foster–Vohra calibration algorithm (Foster & Vohra, 1998) drives empirical calibration error to zero in the Cesàro sense via a Kakutani-type fixed-point construction. Because for Bernoulli prediction any per-round action $p \in [0, 1]$ can be rationalized as a Bayes act under *some* prior with mean $p$, the literal claim "not Bayes for any prior" is uninformative. The substantive distinction is that the Foster–Vohra sequence is not sequentially Bayes coherent under a single prior predictive law (C2 of Definition 4.1); we do not rely on this in the separation proof above.)

*(iii) $\mathfrak{A} \not\subseteq \mathfrak{D}$.* Take the explicit predictive system $\Delta = (\delta^0, E^{\mathrm{LR}}, \hat{C}^{\mathrm{conf}}, \sigma^0)$ where $\delta^0 \equiv 0$ and $\sigma^0 \equiv 0$ are the constant zero predictor and online strategy. The e-process $E^{\mathrm{LR}}$ is a nonnegative martingale under $\mathcal{H}_0$, hence AV-admissible (Theorem 5.6), so $\Delta \in \mathfrak{A}$. The strategy $\sigma^0$ is not CApp boundary-feasible under log loss: $R(\theta, 0) = +\infty$ for every $\theta \in (0, 1)$, so $\bar{R}_n(\theta, \sigma^0) = +\infty \not\to H(\theta)$. Hence $\Delta \notin \mathfrak{D}$.

*(iv) $\mathfrak{D} \not\subseteq \mathfrak{A}$.* Take $\Delta = (\delta^{\mathrm{pi}}, E^\lambda, \hat{C}^{\mathrm{conf}}, \sigma^{\mathrm{KT}})$ with $E_n^\lambda = \lambda^n E_n^{\mathrm{LR}}$, $\lambda \in (0, 1)$. The strategy $\sigma^{\mathrm{KT}}$ is CApp boundary-feasible by Lemma 6.9, so $\Delta \in \mathfrak{D}$. The e-process $E^\lambda$ is a strict supermartingale in $\mathcal{C}_{\mathrm{AV}}$, hence inadmissible by Theorem 5.6 (strictly dominated by $E^{\mathrm{LR}}$), so $\Delta \notin \mathfrak{A}$. (The constant unit process $E \equiv 1$ is itself a nonnegative martingale and therefore AV-admissible, which is why we use the strict supermartingale $E^\lambda$ as the failure witness.)

*(v) $\mathfrak{C} \not\subseteq \mathfrak{D}$.* Take $\Delta = (\delta^0, E^{\mathrm{LR}}, \hat{C}^{\mathrm{conf}}, \sigma^0)$ as in case (iii). The conformal set $\hat{C}^{\mathrm{conf}}$ is coverage-admissible within $\mathcal{G}_s$ (Proposition 5.14), so $\Delta \in \mathfrak{C}$; the strategy $\sigma^0 \equiv 0$ has $\bar{R}_n(\theta, \sigma^0) = +\infty$ under log loss, so $\Delta \notin \mathfrak{D}$.

*(vi) $\mathfrak{D} \not\subseteq \mathfrak{C}$.* Take $\Delta = (\delta^{\mathrm{pi}}, E^{\mathrm{LR}}, \hat{C}^{(q')}, \sigma^{\mathrm{KT}})$ with threshold $q' > \hat{q}_{1-\alpha}$ in the same score-threshold family $\mathcal{G}_s$ as $\hat{C}^{\mathrm{conf}} = \hat{C}^{(\hat{q}_{1-\alpha})}$. Then $\sigma^{\mathrm{KT}} \in \mathfrak{D}$ as in case (iv); the threshold $q'$ is chosen so that the enlargement is strict in $\mathcal{G}_s$ (strictly larger expected length at some $n$), so $\hat{C}^{(q')}$ is dominated within $\mathcal{G}_s$ by $\hat{C}^{(\hat{q}_{1-\alpha})}$ and is therefore not coverage-admissible. Both sets remain in $\mathcal{F}_{\mathrm{Cov}}(\alpha)$, so the failure is a substantive dominance failure in the set coordinate, and $\Delta \notin \mathfrak{C}$. $\qquad\square$

## 6.6 Interpretation: four irreducible geometries

The extended separation theorem identifies four conceptions of statistical optimality:

(i) *Blackwell*: the rule is optimal for a declared objective (prior-witnessed, per-round);
(ii) *Anytime-valid*: the rule controls error at every stopping time (martingale-witnessed);
(iii) *Coverage*: the rule guarantees marginal containment at level $1 - \alpha$ with minimal expected length (exchangeability-witnessed);
(iv) *CApp*: the rule reaches the boundary in the long run (fixed-point-witnessed, limiting).

Each geometry defines its own dominance order on its own ambient space of procedures; a witness that is admissible under one order may fail another, as Theorems 5.21–6.12 establish. A certificate in one geometry is not a certificate in the other three. Table 1 summarizes the four geometries.

Table 1: Taxonomy of four admissibility geometries.

| Geometry | Certificate | Boundary witness | Optimality mode |
|---|---|---|---|
| $\mathfrak{B}$: Blackwell | supporting hyperplane | prior $\Pi$ | pointwise (per-round) |
| $\mathfrak{A}$: Anytime-valid | supermartingale | e-process | pathwise (all stopping times) |
| $\mathfrak{C}$: Coverage | feasibility region | conformal rank | marginal (exchangeability) |
| $\mathfrak{D}$: CApp | Cesàro steering | fixed-point / minimax | Cesàro (time-averaged) |

The four-geometry diamond (Figure 5) illustrates the pairwise non-nesting. No arrow connects any pair, reflecting the non-nesting established in Theorems 5.21–6.12.

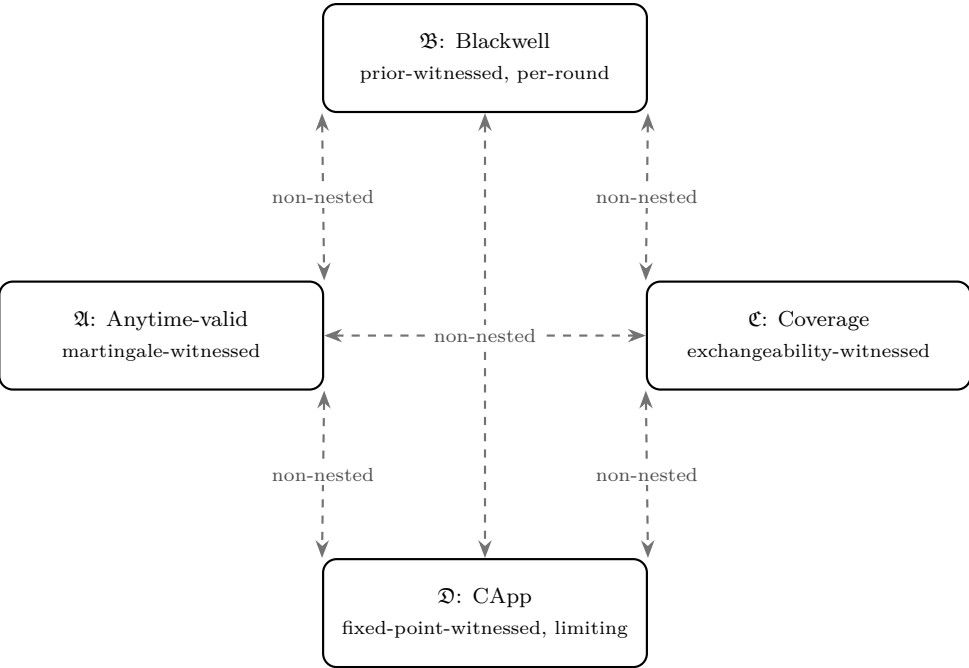

Figure 5: Four admissibility geometries in diamond configuration. Each node represents an admissible class; dashed arrows indicate pairwise non-nesting (Theorems 5.21 and 6.12). Blackwell admissibility and CApp boundary-feasibility share the risk-set domain but differ in witness type (prior vs. fixed-point); anytime-valid and coverage admissibility operate on different procedure spaces entirely.

## 7 Bernoulli Laboratory

The Bernoulli model is a minimal laboratory for predictive inference: small enough to compute in closed form, expressive enough to exhibit all four geometries. The separation theorem is proved constructively using four procedures in the Bernoulli model. This section collects the classification results in a single table and schematic, making the pairwise non-nesting visually explicit.

Table 2 records the four procedures and four admissibility criteria. N/A indicates that the criterion is defined for a categorically different class of procedures; the assessment is inapplicable rather than negative.

**Corollary 7.1** (Martingale property is not criterion-determining). *P1, P2, P3 all satisfy the martingale property and belong to different admissibility classes. The martingale property does not determine membership in $\mathfrak{B}$, $\mathfrak{A}$, or $\mathfrak{C}$.*

**Corollary 7.2** (No universal certificate). *No one of the four admissibility certificates (Blackwell supporting-hyperplane, anytime-valid martingale, fixed-level coverage frontier in expected length, Cesàro steering) implies*

Table 2: Bernoulli laboratory: four procedures and four criterion classes. Column headers: *Martingale property* under the appropriate measure (Definition 4.1); *Blackwell admissibility*; *AV-admissibility* (in the sense of Theorem 5.6); *Coverage validity* (at level $1 - \alpha$, Definition 5.12); *CApp boundary-feasibility* (Definition 6.7). $\checkmark$ = satisfies; $\times$ = fails; N/A = criterion not applicable to this procedure type. Cross-type cells are N/A rather than $\times$: a criterion that is undefined for a procedure type is not thereby refuted. Substantive cross-criterion failures require active coordinates and are established in Theorems 5.21 and 6.12. The unclipped plug-in MLE assigns zero probability to realizable events whenever $S_n \in \{0, n\}$, giving infinite log loss; its Cesàro-average risk does not converge to $H(\theta)$ in the extended-real sense, so it is not CApp boundary-feasible under log loss. [†]Clipping by $\epsilon > 0$ removes the boundary blowup, so the clipped predictor has finite risk and its time-averaged log loss approaches $H(\theta)$ at rate $O((\log n)/n)$. We include the clipped row only as a numerical convention (Section 8) and make no finite-sample admissibility claim about it.

| Procedure | Martingale property | Blackwell admissible | AV-admissible | Coverage valid | CApp feasible |
|---|---|---|---|---|---|
| P1: Bayes shrinkage / KT-type predictor | $\checkmark$ | $\checkmark$ | N/A | N/A | $\checkmark$ |
| P2: Plug-in MLE $S_n/n$ (unclipped) | $\checkmark$ | $\times$ | N/A | N/A | $\times$ |
| P2$^\epsilon$: Clipped plug-in (numerical convention) | $\checkmark$ | not used | N/A | N/A | $\checkmark^\dagger$ |
| P3: LR e-process | $\checkmark$ | N/A | $\checkmark$ | N/A | N/A |
| P4: Conformal prediction set | N/A | N/A | N/A | $\checkmark$ | N/A |

*Note.* On the full Bernoulli model, the Jeffreys Bayes predictive is the KT closed form $(S_n + \frac{1}{2})/(n + 1)$, whose CApp boundary-feasibility is established by Lemma 6.9. On compact submodels $\Theta = [\eta, 1 - \eta] \subset (0, 1)$, the Blackwell-admissible witness is the restricted full-support Bayes predictive $\delta_n^B = \mathbb{E}_{\Pi_J}[\theta \mid X_{1:n}]$ under the renormalized Jeffreys prior, certified by Corollary 3.14; this is generally not the KT closed form.

*all the others. A modular predictive system in $\Sigma$ may satisfy several of the four criteria componentwise (examples are recorded in Section 5.7), but joint admissibility must be specified one coordinate at a time, and cannot be obtained from a single one of the four orders.*

## 7.1 Extended procedures

The extended proof of Theorem 6.12 uses two additional ingredients: the Krichevsky–Trofimov forecaster $\hat{p}_n^{\mathrm{KT}} = (S_n + \frac{1}{2})/(n + 1)$ as a CApp boundary-feasible strategy under log loss (Lemma 6.9), and a misspecified-prior Bayes rule $\delta_\Pi \equiv 0.3$ on the compact Bernoulli model $\Theta = [\eta, 1 - \eta] \ni 0.3, 0.7$ as a Blackwell-admissible predictor whose Cesàro-average risk at the true parameter never reaches the entropy floor. Foster–Vohra calibration (Foster & Vohra, 1998) is discussed only as an interpretive example under a calibration-error objective, not as a log-loss CApp witness (Remark 6.10).

**Beyond Bernoulli.** The Bernoulli laboratory is deliberately minimal; analogous witnesses can be constructed in Gaussian and nonparametric predictive models (e.g. the sample mean and a likelihood-ratio $e$-process in the Gaussian location family, or posterior predictives under a Dirichlet-process prior). Developing the complete-class details in those settings would require additional topological machinery and is beyond the present paper; the non-nesting argument carries over with only notational changes to the sample space and likelihood structure.

## 8 Numerical Illustrations

The preceding sections established the separation theorem analytically. We now illustrate each geometry in finite samples, to show how the distinctions appear in small simulations. Each of the following three Monte Carlo experiments targets one admissibility geometry; all use $B = 10,000$ replications.

## 8.1 Bayes vs. plug-in under log loss

We draw $X_1, \ldots, X_n \stackrel{\text{iid}}{\sim} \text{Bern}(0.3)$ and compare the next-step log loss $L(\theta, p) = -\theta \log p - (1 - \theta) \log(1 - p)$ for the Bayes predictive $\hat{p}_n^B = (S_n + \frac{1}{2})/(n + 1)$ under $\text{Beta}(\frac{1}{2}, \frac{1}{2})$ and the plug-in MLE $\hat{p}_n^{\text{pi}} = S_n/n$. Theorem 4.4(iii) states that the unclipped MLE has *infinite extended-real risk for every finite $n$*, because $P_\theta(S_n \in \{0, n\}) = (1 - \theta)^n + \theta^n$ is strictly positive for every $\theta \in (0, 1)$. The simulation therefore reports two distinct quantities (Table 3): a numerically clipped MLE risk (bounding the MLE away from the simplex boundary by $\epsilon = 10^{-6}$), and the *theoretical* unclipped risk, which is $+\infty$ for every $n$. The clipped column is a *numerical convention* only, included for visual comparison with the Bayes column.

Table 3: Average log loss in nats ($\theta = 0.3$, $B = 10{,}000$). The unclipped MLE is $+\infty$ on every realization where $S_n \in \{0, n\}$, an event with strictly positive probability $(1 - \theta)^n + \theta^n > 0$ for every finite $n$ and every $\theta \in (0, 1)$; the theoretical unclipped MLE risk is therefore $+\infty$ uniformly in $n$. The clipped column applies the convention $\hat{p}_n^{\text{pi}} \mapsto \max(\epsilon, \min(1 - \epsilon, \hat{p}_n^{\text{pi}}))$ with $\epsilon = 10^{-6}$ before computing log loss; this is a numerical convention only and should not be interpreted as the true risk. The final column reports the exact theoretical boundary probability $P_\theta(S_n \in \{0, n\}) = 0.7^n + 0.3^n$.

| $n$ | Bayes risk | MLE clipped ($\epsilon = 10^{-6}$) | MLE unclipped (theoretical) | $P_\theta\{S_n \in \{0, n\}\}$ |
|-----|-----------|-----------------------------------|----------------------------|-------------------------------|
| 5 | 0.694 | 1.286 | $+\infty$ | $1.70 \times 10^{-1}$ |
| 10 | 0.658 | 0.757 | $+\infty$ | $2.83 \times 10^{-2}$ |
| 25 | 0.630 | 0.633 | $+\infty$ | $1.34 \times 10^{-4}$ |
| 50 | 0.621 | 0.621 | $+\infty$ | $1.80 \times 10^{-8}$ |
| 100 | 0.616 | 0.616 | $+\infty$ | $3.23 \times 10^{-16}$ |

The boundary probability shrinks rapidly with $n$ but is never zero: $(0.7)^{100} + (0.3)^{100} \approx 3.23 \times 10^{-16}$ is positive though far below any Monte Carlo resolution. The clipped MLE numerically converges to the Bayes risk because the boundary event becomes unobservable in samples of size $B$, but the *theoretical* risk of the unclipped MLE remains $+\infty$ for every $n$. The Bayes rule, by contrast, has finite risk uniformly across all realizations and is never dominated for any $n \geq 1$, consistent with Theorem 4.4 and the risk-set geometry of Section 3.

## 8.2 Anytime validity: e-process vs. naive peeking

We draw $X_1, \ldots, X_{200} \stackrel{\text{iid}}{\sim} \text{Bern}(0.5)$ under $H_0 : \theta = 0.5$ and compare two sequential testing strategies at nominal level $\alpha = 0.05$. The *e-process* accumulates a running likelihood ratio using the Bayes predictive as the alternative plug-in and rejects whenever $E_t \geq 1/\alpha$; it is an element of $\mathcal{C}_{\text{AV}}$ and controls type-I error at every stopping time (Theorem 5.6). The *naive strategy* performs a $z$-test at each of five pre-specified sample sizes $n \in \{10, 20, 50, 100, 200\}$ and rejects if any test exceeds $z_{0.025} = 1.96$. Table 4 records the rejection rates. The e-process remains below $\alpha$ while naive peeking inflates the type-I error to 0.165, showing that $\mathcal{C}_{\text{AV}}$-feasibility is a binding constraint that the unrestricted testing class violates.

Table 4: Type-I error under $H_0 : \theta = 0.5$ ($B = 10{,}000$, $\alpha = 0.05$).

| Strategy | Rejection rate |
|----------|---------------|
| E-process (anytime-valid) | 0.031 |
| Naive peeking (5 looks) | 0.165 |
| Nominal $\alpha$ | 0.050 |

## 8.3 Conformal coverage under covariate shift

We set $Y \mid X = x \sim N(0, (1 + x)^2)$ and construct split-conformal prediction intervals using the naive score $s(x, y) = |y|$ with $n_{\text{cal}} = 500$ calibration points and $n_{\text{test}} = 2{,}000$ test points at level $1 - \alpha = 0.90$. Table 5

compares three scenarios: (A) calibration and test both under $X \sim \text{Uniform}[0, 1]$; (B) calibration under Uniform$[0, 1]$, test under $X \sim \text{Beta}(2, 5)$; (C) both under Beta$(2, 5)$. When calibration and test distributions match, marginal coverage holds at the nominal level. Under covariate shift (Scenario B), the Uniform-calibrated quantile is too wide for the Beta$(2, 5)$ test population (which concentrates $X$ near zero, where the conditional variance is smaller), inflating coverage to 0.946. Re-calibrating under the test distribution (Scenario C) restores nominal coverage and yields a tighter interval. The calibration quantile itself shifts from 2.51 to 2.13, illustrating that marginal coverage validity is a property of the exchangeable joint distribution, not a universal guarantee across arbitrary design points, consistent with Theorem 5.16 and the scope of $\mathcal{C}_{\text{Cov}}$ in Definition 5.11. Scenario B breaks exchangeability, so the conformal guarantee no longer applies; the observed overcoverage reflects distributional mismatch, not a violation or strengthening of the theorem.

Table 5: Split-conformal coverage ($1 - \alpha = 0.90$, $n_{\text{cal}} = 500$, $n_{\text{test}} = 2{,}000$).

|   | Calibration $\rightarrow$ Test | Quantile | Coverage | Half-width |
|---|---|---|---|---|
| A | Unif $\rightarrow$ Unif | 2.51 | 0.900 | 2.51 |
| B | Unif $\rightarrow$ Beta$(2, 5)$ | 2.51 | 0.946 | 2.51 |
| C | Beta$(2, 5) \rightarrow$ Beta$(2, 5)$ | 2.13 | 0.900 | 2.13 |

## 9 Motivations and interpretations

We close by sketching three areas in which the criterion-relative viewpoint of this paper provides a useful interpretive lens. The discussions in this section are *design interpretations* of the criterion-relative framework rather than new theorem statements. They illustrate how the four certificates appear in modern predictive systems.

**How to use this paper.** For practitioners building predictive systems that must satisfy several criteria at once, the framework suggests a certificate checklist. *First*, identify which guarantee is required: a Blackwell certificate (a supporting-hyperplane prior under the declared loss), an anytime-valid certificate (a nonnegative martingale controlling error at all stopping times), a coverage certificate (an exchangeability rank together with expected-length efficiency inside the chosen score family), or a CApp certificate (a Cesàro steering or regret argument under the chosen loss). *Second*, attach each certificate to the appropriate coordinate of a predictive system $\Delta \in \Sigma$ via the constrained Bayes template (Definition 5.18): specify $(\mathcal{D}_C, \mathcal{F}_C, \mathcal{R}_C)$ and then optimize Bayesian integrated risk within $\mathcal{F}_C$ (Figure 6). A modular system may carry several certificates, but none automatically transports to the others: a conformal wrapper does not make the base predictor Blackwell admissible, a calibrated forecaster need not be log-loss optimal, and an anytime-valid monitor need not be fixed-sample most powerful.

For example, a foundation-model pipeline may combine a log-loss-trained predictor, a sequential safety monitor, a conformal uncertainty wrapper, and a calibration audit; the point of $\Sigma$ is to record which certificate belongs to which component.

### 9.1 Probabilistic forecasting and calibration

Calibration in the empirical sense (predicted probabilities matching observed frequencies in time average) (Dawid, 1982; Foster & Vohra, 1998) is property C3 of Definition 4.1: self-consistency under the forecaster's own predictive measure. The Bernoulli plug-in counterexample (Theorem 4.4) shows that this certificate does not transport to Blackwell admissibility under log loss: a calibrated forecaster may still be dominated by a Bayes-shrunk predictor. The constrained Bayes schema (Definition 5.18) suggests a design heuristic: if calibration is required, treat it as a feasibility constraint and optimize Bayesian log loss inside the calibration-feasible class. We do not prove a new admissibility theorem in this geometry.

## 9.2 Sequential testing and safe monitoring

In sequential testing, the AV-admissibility geometry corresponds to type-I error control at every data-dependent stopping time, whereas Neyman–Pearson optimality corresponds to power maximization at a fixed sample size. Theorem 5.21 shows that these two criteria operate on different decision spaces under different partial orders: an AV-admissible $e$-process need not be the most powerful fixed-sample test, and a Neyman–Pearson optimal test need not be AV-admissible. Within AV, the constrained Bayes schema selects a prior or mixture to optimize expected growth and power inside $\mathcal{C}_{\text{AV}}$, recovering the growth-rate optimal $e$-value (GROW) construction of Grünwald et al. (2024) as an instance of the schema. The contribution here is organizational: fixed-sample power and anytime-valid safety are different certificates.

## 9.3 Conformal wrappers for black-box predictors

A conformal wrapper produces a predictive system in $\Sigma$ whose set coordinate $\hat{C}^{\text{conf}}$ lies in $\mathfrak{C}$ (expected-length efficient within its score-threshold family in $\mathcal{F}_{\text{Cov}}(\alpha)$; Proposition 5.14), while the predictor coordinate may or may not lie in $\mathfrak{B}$. Thus a conformal wrapper transports a *coverage* certificate to the system, not a Blackwell certificate to the base model. Conversely, a Bayes or log-loss-optimal predictor does not by itself supply marginal coverage. This is the meta-space picture in its most common machine-learning form; see Angelopoulos & Bates (2023) for a survey of conformal wrappers in current practice. The conformal impossibility result of Foygel Barber et al. (2021) (Theorem 5.16) remains the binding constraint on conditional coverage.

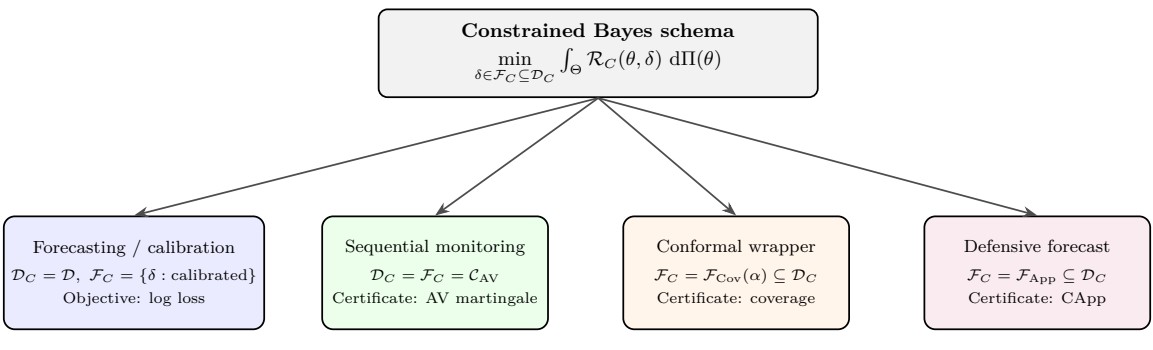

*Specify $(\mathcal{D}_C, \mathcal{F}_C, \mathcal{R}_C)$ first, then optimize Bayesian risk within $\mathcal{F}_C$.*

Figure 6: The constrained Bayes schema applied to four domains. In each case, the criterion $C$ supplies its own decision space $\mathcal{D}_C$, feasibility set $\mathcal{F}_C \subseteq \mathcal{D}_C$, and risk functional $\mathcal{R}_C$; the optimization objective is Bayesian integrated risk within $\mathcal{F}_C$ (Definition 5.18 and Remark 5.19). The resulting procedure is admissible relative to its criterion but not necessarily admissible under the other three (Theorems 5.21 and 6.12).

## 10 Discussion

The central result is that admissibility is criterion-relative. We do not argue that any one framework is superior; rather, different inferential guarantees correspond to different objects, orders, and certificates. Because point predictors, $e$-processes, prediction sets, and online strategies inhabit different coordinates of $\Sigma$, their criterion classes cannot be collapsed into a single criterion-neutral admissibility order. The criterion separation theorem (Theorem 5.21) and its extension (Theorem 6.12) show these classes are pairwise non-nested, and the Bernoulli laboratory makes the non-nesting constructive. Within the Blackwell framework, Proposition 3.7 and Theorem 3.8 identify no-shame rules with lower-boundary risk points supported by a prior; Corollaries 3.12 and 3.14 give the full-support conditions (finite and compact $\Theta$, respectively) under which Bayes rules are guaranteed no-shame. The resulting picture is not relativism but *disciplined pluralism*: once a certificate is chosen, its dominance order is binding, but no certificate supplies all the others.

Martingale coherence links single-prior Bayes prediction and anytime-valid inference, but the link is measure-relative and does not extend to coverage or CApp. The plug-in MLE example, while small, illustrates a

pattern that surfaces informally in modern probabilistic forecasting: self-consistency under the forecaster's own predictive distribution is not the same as non-dominance under the true data-generating process. We have framed the discussions of forecasting, sequential testing, and conformal wrappers in Section 9 as design interpretations through the criterion-relative lens; they motivate the framework and illustrate how it organizes existing recipes, but they are not consequences of the results proved here.

The constrained Bayes formulation (Definition 5.18) provides a common design template for all four criteria. It is not a unification: the four criteria operate on different decision spaces and induce different partial orders. The template makes the criterion-relative design choice explicit (specify the validity constraint first, then optimize Bayesian risk within it) but does not produce a master ordering, as Theorems 5.21 and 6.12 make precise. Geometrically, $\Pi$ acts as a supporting-hyperplane normal selecting an exposed point of the restricted risk frontier within each criterion separately (Remark 5.20).

The constructive-versus-Cesàro distinction (Section 6.1) clarifies why prior-witnessed Bayes procedures and choice-based calibration or approachability procedures supply different certificates. Under log loss our CApp witness is the KT forecaster (Lemma 6.9); under calibration-error objectives, defensive forecasting (Vovk et al., 2005b) and Foster–Vohra (Foster & Vohra, 1998) illustrate the choice-based route through fixed-point arguments rather than prior witnesses.

Several directions remain open. Extending the separation theorem to nonparametric and infinite-dimensional models would clarify the scope of the $\Sigma$-space construction. Understanding how constrained Bayes interacts with minimax phenomena, especially in high-dimensional regimes where James–Stein-type effects arise, would connect the framework to classical complete-class theory. Finally, modern ML pipelines raise a practical multi-objective question: how should one choose, combine, and communicate certificates when calibration, coverage, and sequential validity are all desired?

**Broader-impact remark.** Criterion-relativity should not weaken application-specific accountability. In deployed predictive systems, practitioners should state which certificate is being used, why it is appropriate, and which guarantees are *not* provided. A system satisfying one criterion does not inherit the guarantees of the others.

The practical lesson is therefore simple: state the certificate, state the coordinate, and do not transport guarantees across geometries without an additional argument.

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
