# OpenReview forum: "Bayes with No Shame: Admissibility Geometries of Predictive Inference"
_TMLR — Decision pending for TMLR_

### Review · Reviewer_CP4U · 2026-05-20

**Summary Of Contributions:**

The paper proposes a unifying geometric discussion of several notions of admissibility and optimality in predictive inference. It contrasts Blackwell risk admissibility, anytime-valid admissibility for e-processes, marginal coverage validity in conformal prediction, and a Cesàro/approachability-based notion of long-run admissibility. The main claim is that these four criteria define distinct “admissibility geometries” and that the corresponding classes of procedures are pairwise non-nested. The paper also emphasizes the role, and the limitations, of martingale coherence: martingale structure is central for Bayesian posterior predictives and e-processes, but is argued not to provide a universal notion of admissibility.

A strength of the paper is that it addresses an interesting conceptual issue, since different communities use different notions of optimality, validity, and calibration, and these notions should not be conflated. The Bernoulli plug-in versus Bayes example under log loss is a useful illustration of the difference between self-consistency and decision-theoretic admissibility.

However, the main weaknesses are substantial. The separation results appear to rely heavily on comparing procedures of different mathematical types, rather than establishing non-nesting within a common decision space. Some notions called “admissibility”, especially marginal coverage validity, are closer to feasibility constraints than to genuine dominance-based admissibility concepts. Several technical claims also seem to require stronger assumptions or more careful formulation. As a result, while the paper raises an interesting conceptual point, I do not find the main mathematical contribution convincingly established in its current form.

**Audience:**

Yes

**Audience Explanation:**

I think some readers in the TMLR audience would be interested in the paper’s general message. The manuscript addresses an important issue for machine learning and statistical inference: different communities use different notions of optimality, validity, calibration, and admissibility, and these notions should not be conflated. The attempt to relate Blackwell admissibility, anytime-valid inference, conformal prediction, and approachability-style calibration could be useful to readers working on uncertainty quantification, sequential testing, calibration, and probabilistic prediction.

The Bernoulli plug-in versus Bayes example under log loss is also a useful pedagogical illustration of the distinction between self-consistency or calibration-like behavior and decision-theoretic admissibility.

However, I do not think this potential interest is sufficient to compensate for the paper’s technical weaknesses. In particular, the main separation results are not convincingly established, because they rely heavily on comparing procedures of different mathematical types, and several central notions of admissibility are not defined with enough precision. Thus, while the topic and perspective may be of interest to some TMLR readers, the findings in their current form are not sufficiently reliable.

**Broader Impact Concerns:**

I do not see major direct ethical risks, since the paper is primarily theoretical. However, because the paper discusses evaluation criteria for deployed predictive systems, the authors should make clear that criterion-relativity does not weaken the need for application-specific accountability. Practitioners should still state which validity criterion is used, why it is appropriate, and which guarantees are not provided.

**Claims And Evidence:**

No

**Claims Explanation:**

I do not think the main claims are supported by sufficiently accurate, convincing, and clear evidence. The main issue is that the central “criterion separation” theorem is not formulated in a common mathematical universe. The paper denotes by $B,A,C,D$ the classes of Blackwell-admissible, anytime-valid admissible, coverage-valid, and CAA-admissible procedures, but these classes are not subsets of the same ambient space. Roughly, $B \subseteq \mathcal D_{\rm pred}$, $A \subseteq \mathcal D_{\rm eproc}$, $C \subseteq \mathcal D_{\rm set}$, and $D \subseteq \mathcal D_{\rm seq}$. Thus, statements such as $B\not\subseteq A$ or $A\not\subseteq C$ are not very meaningful unless all four classes are first embedded into a common comparison space. In the proof of Theorem 5.9, the witness procedures fail the other criteria mostly because they are objects of the wrong type: a point predictor is not an e-process, an e-process is not a prediction set, and a conformal prediction set is not a point predictor. This establishes categorical inapplicability, but not a substantive non-nesting theorem.

A second issue is that marginal coverage is treated as an admissibility notion, whereas the paper defines only a validity constraint: $ \mathbb P\{Y_{n+1}\in \widehat C_n(X_{n+1})\}\ge 1-\alpha $. This is a feasibility condition, not an admissibility order. If the partial order is only by coverage level, then the trivial prediction set $\widehat C_n(x)=\mathcal Y$ dominates most other sets in coverage. If the intended notion is instead efficiency on the coverage frontier, then the paper would need a dominance relation involving length, volume, or sharpness, for example $\mathrm{Cov}(\widehat C')\ge \mathrm{Cov}(\widehat C)$ and $\mathbb E|\widehat C'|\le \mathbb E|\widehat C|$, with at least one strict inequality. Such an admissibility notion is not clearly defined.

Several technical claims also seem to require stronger assumptions. For instance, the statement that every Bayes rule is “no-shame” is too strong under the stated compact-$\Theta$ framework. If a rule $\delta'$ improves on a Bayes rule $\delta_\Pi$ only at a point $\theta_0$ with $\Pi(\{\theta_0\})=0$, then $R(\theta,\delta')\le R(\theta,\delta_\Pi)$ for all $\theta$, with $R(\theta_0,\delta')<R(\theta_0,\delta_\Pi)$, does not imply $\int R(\theta,\delta')\,d\Pi(\theta) < \int R(\theta,\delta_\Pi)\,d\Pi(\theta)$. The proof would be correct for finite $\Theta$ with strictly positive prior weights, or under additional continuity/support assumptions, but not as stated.

Similarly, the claimed implication from constructive admissibility to martingale coherence appears unjustified. The definition allows a prior $\Pi_n$ at each sample size $n$. But being Bayes at each $n$ for a possibly different prior does not imply the existence of a single prior predictive law under which the sequence is a martingale. Martingale coherence would require something like $\widehat p_n = \mathbb E_\Pi[\theta\mid\mathcal F_n]$ for one fixed prior $\Pi$, not merely pointwise Bayes rationalizability by a sequence $(\Pi_n)_n$.

Finally, the extended CAA separation theorem is not convincingly proved. The proof of $B\not\subseteq D$ seems to argue that a Bayes procedure is not “defined by” an approachability condition, but this does not show that it fails the CAA property. Conversely, the claim that defensive forecasting drives time-averaged risk to the lower boundary of the risk set is asserted rather than established. Long-run calibration and convergence to the Blackwell lower risk boundary under a specified loss are distinct properties and should not be conflated.

Overall, the paper raises an interesting conceptual distinction between several inferential paradigms, but the main mathematical claims are not supported by sufficiently precise definitions and convincing proofs.

**Requested Changes:**

1. Critical: Reformulate the separation theorems in a common space.

The current statements that $B,A,C,D$ are pairwise non-nested are not convincing because these classes are not defined as subsets of the same mathematical universe. The proof often shows that a procedure fails another criterion simply because it is an object of the wrong type: a point predictor is not an e-process, an e-process is not a prediction set, and a prediction set is not a point predictor. The authors should either:

(a) define a common meta-space of procedures, for example procedures with several components such as a predictor, an e-process, a prediction set, and an online strategy, and then define $B,A,C,D$ as properties within that common space; or

(b) substantially weaken the claim and present the result as a taxonomy of different evaluation paradigms, rather than as a pairwise non-nesting theorem.

Without such a reformulation, the main theorem remains largely categorical rather than a substantive mathematical separation result.

2. Critical: Define genuine admissibility notions for the non-Blackwell criteria.

The paper repeatedly describes marginal coverage as an admissibility geometry, but the formal definition is only the feasibility constraint $ \mathbb P\{Y_{n+1}\in \widehat C_n(X_{n+1})\}\ge 1-\alpha $. This is not, by itself, an admissibility relation. If the order is only by coverage, then the trivial set $\widehat C_n(x)=\mathcal Y$ dominates essentially everything. The authors should define a proper dominance relation for prediction sets, for instance involving both coverage and expected length/volume/sharpness, and then define admissibility on the corresponding frontier.

Analogous care is needed for the CAA criterion. The paper should specify exactly what the ambient space is, what the order is, and what it means for one sequential strategy to dominate another.

3. Critical: Repair or remove Theorem 5.9 and Theorem 6.7.

The proof of Theorem 5.9 currently establishes mostly structural inapplicability across object types. The proof of Theorem 6.7 is even less convincing. For example, the argument for $B\not\subseteq D$ appears to say that a Bayes procedure is not “defined by” an approachability condition, but this does not prove that it fails the CAA property. A procedure can satisfy a property even if it was not constructed for that purpose.

Similarly, the claim that defensive forecasting is in $D$ requires a real proof that its time-averaged risk converges to the lower boundary $\partial_-R$ for the relevant loss and model. Long-run calibration is not automatically the same as convergence to the Blackwell lower risk boundary.

The separation claims should either be fully proved under precise definitions, or weakened to informal/non-theorem-level discussion.

4. Critical: Correct the claims relating Bayes rules and Blackwell admissibility.

Statements such as “every Bayes rule is no-shame” require stronger assumptions than those currently stated. If $\Theta$ is finite and the prior assigns positive mass to every $\theta$, the argument is standard. But for compact $\Theta$ with general priors, a strict improvement at a point $\theta_0$ with $\Pi(\{\theta_0\})=0$ need not reduce the integrated Bayes risk. The paper should add the required support/continuity assumptions, or restrict the result to the finite-parameter setting.

5. Critical: Strengthen the supporting-hyperplane argument.

The proof of Theorem 3.8 is too quick. In particular, the argument for the nonnegativity of the supporting vector $\pi$ is not valid as written: one cannot generally decrease one coordinate of a risk vector while holding the others fixed and remain inside the risk set. The authors should replace this with a standard convex-analysis argument for Pareto-efficient points of a closed convex risk set, and clearly state the closedness/topological assumptions required.

6. Critical: Fix Proposition 6.4.

Constructive admissibility is defined using a prior $\Pi_n$ at each sample size $n$. But being Bayes at each $n$ for a possibly different prior does not imply martingale coherence. Martingale coherence would require a single prior predictive law, or at least a compatible family of priors, such that $\widehat p_n=\mathbb E_\Pi[\theta\mid\mathcal F_n]$. The proposition should either be corrected with this additional assumption or removed.

7. Critical: Clarify the role of martingale coherence.

The paper sometimes suggests that martingale coherence is necessary for Blackwell admissibility. What is clearly shown is that Bayesian posterior predictives are martingales under the prior predictive law, and that the plug-in example shows martingale-like self-consistency is not sufficient for Blackwell admissibility. These are different claims. The authors should distinguish carefully between fixed-sample admissibility, sequential Bayesian coherence, and self-consistency under a data-dependent predictive law.

8. Critical: Make the experiments consistent with the theory.

The theory says that the Bernoulli plug-in MLE has infinite log-loss risk whenever boundary predictions occur with positive probability. However, the simulation table reports finite MLE risks. If the simulations use clipping or another numerical regularization, this must be stated explicitly. Otherwise, the experiment does not illustrate the extended-real risk theorem as claimed.

9. Would strengthen the paper: Tone down the “unifying” claims.

The constrained Bayes schema is interesting as a conceptual device, but it currently risks overstating the unity of the four frameworks. Since the decision spaces, orders, and certificates differ, the paper should be clearer that this is a schematic analogy rather than a single optimization theorem covering all cases.

10. Would strengthen the paper: Separate conceptual contribution from formal theorem statements.

The paper has a potentially valuable message: different communities use different notions of validity and optimality, and these should not be conflated. This message would be stronger if the authors clearly separated it from formal claims that are not fully established.

11. Would strengthen the paper: Rework or shorten the philosophical language.

Terms such as “no-shame”, “moral pluralism”, and the analogy with Berlin/Williams may be useful rhetorically, but they sometimes obscure the mathematical content. The paper would be clearer if the main text focused on the precise statistical and decision-theoretic distinctions, with philosophical remarks moved to a short discussion.

12. Would strengthen the paper: Reconsider the conjectural Baire-category appendix.

The appendix on meagerness/universal admissibility is interesting but highly speculative and relies on additional topological assumptions. Since it is not central to the main contribution, it should either be removed, clearly labeled as speculative, or developed with rigorous assumptions.

---

> ### Author Response · Authors · 2026-05-20
> **Acknowledgments**
>
> Thank you for the careful and mathematically detailed review. We greatly appreciate the reviewer’s close engagement with both the conceptual and technical aspects of the manuscript.
>
> We agree that several parts of the paper would benefit from sharper formulation and clarification, particularly regarding:
>
> - the distinction between validity constraints and admissibility relations,
> - the role of differing inferential object types and their associated ambient spaces,
> - and the precise scope of the separation theorems.
>
> We also appreciate the reviewer’s comments concerning the supporting-hyperplane argument, the martingale coherence claims, and the assumptions underlying the Bayes admissibility results. Several of these observations are well taken and will help us improve both the precision and presentation of the manuscript.
>
> At the same time, we found the reviewer’s remarks on typed procedure spaces and criterion-relative admissibility especially constructive and insightful. We believe these comments point toward a clearer and more disciplined framing of the paper’s central contribution as a comparative geometry and taxonomy of inferential evaluation regimes.
>
> We are reviewing the comments carefully and expect the revision to include clarified definitions, refined theorem statements, and a sharper separation between conceptual interpretation and formal claims, particularly after consideration of the remaining reviews.
>
> Thank you again for the thoughtful and rigorous assessment.

---

> ### Author Response · Authors · 2026-06-12
> **Response: revision restructured around your two critical observations**
>
> We are grateful for the depth and precision of this review; it identified real gaps, and the revision is restructured around its two central observations.
>
> Common space (your option (a)): the four criteria are now defined on a meta-space $\Sigma$ of predictive systems $\Delta=(\delta,E,\hat{C},\sigma)$ (Defs. 5.1-5.2). The separation theorems (Thms. 5.19, 6.12) construct witnesses that are active in both the membership and failure coordinates, so every failure is a substantive dominance failure, not categorical inapplicability.
>
> Genuine dominance orders: coverage admissibility at level $1-\alpha$ is feasibility plus expected-length undominatedness (Def. 5.10); the trivial set is now strictly dominated, as your example required. CApp is defined as boundary-feasibility toward the per-parameter oracle envelope, with the loss and mode of convergence pinned down (Def. 6.7), and kept distinct from Pareto-frontier membership.
>
> Proofs: the supporting-hyperplane theorem is reproved via the lower-comprehensive hull $\mathcal{R}+\mathbb{R}^k_+$ (Thm. 3.8); Bayes $\Rightarrow$ admissibility now carries full-support hypotheses (Cors. 3.12, 3.14; asymmetry in Rem. 3.13); Prop. 6.4 is split into single-prior and Kolmogorov-compatible versions; the $\mathfrak{B}\not\subseteq\mathfrak{D}$ case uses an explicit misspecified-prior witness with exact arithmetic, replacing the "not defined by approachability" sketch; the load-bearing log-loss CApp witness is now the KT forecaster, and the defensive-forecasting claim is demoted to a remark (Rem. 6.10).
>
> Coherence and experiments: three coherence notions are declared up front (Def. 4.1) and all "necessary for Blackwell admissibility" phrasing is removed; Table 3 now reports the clipped MLE risk ($\epsilon=10^{-6}$, a declared convention), the theoretical unclipped risk ($+\infty$ uniformly in $n$), and the exact boundary probabilities $(1-\theta)^n+\theta^n$, with the replication script reconciled.
>
> On the would-strengthen items: constrained Bayes is presented as a typed schema, not a unification; contributions are tagged formal/cited/recipe; the philosophical material is compressed to a single disclaimed paragraph in Section 1; the conjectural Baire-category appendix is removed entirely.
>
> The point-by-point response in the supplementary PDF addresses each critical and would-strengthen item with manuscript locations.

---

> > ### Author Response · Authors · 2026-06-21
> > **Revision 2**
> >
> > We have posted a further revision (rev2) with editorial and correctness refinements only; no theorem, proof, table value, or numerical result has changed. Several items sharpen the admissibility definitions and the experiments you scrutinized in the previous round:
> >
> > - the domination order within $\mathcal{C}_{\mathrm{AV}}$ is now stated explicitly ($\succeq_{\mathrm{AV}}$, Definition 5.5), so "AV-admissible" is defined before the Ramdas et al. characterization is invoked;
> > - the complete-class theorem is renamed Wald--Blackwell (matching its cited sources) and stated explicitly under the Section 3 standing-regularity conditions (Theorem 3.10);
> > - the conditional-coverage impossibility is restated precisely as a distribution-free result, and the coverage-admissibility proof now includes the nestedness $\Rightarrow$ monotone-expected-length step (Theorem 5.16, Proposition 5.14);
> > - Table 2 is made internally consistent: the conformal-set martingale entry is now N/A rather than $\times$, and the $\epsilon$-clipped plug-in row is labeled a numerical convention with the unproven finite-$n$ inadmissibility claim removed;
> > - we added the Choe & Ramdas (2026) anytime-valid filtration-transport reference (Remark 5.10).
> >
> > A full itemization with locations is in Section 5 of the response-to-reviewers PDF. Thank you again for the depth of your review.

---

### Review · Reviewer_zCaA · 2026-06-04

**Summary Of Contributions:**

This paper studies whether several active notions of predictive optimality can be understood in single framework. It discusses 4 criteria: Blackwell admissibility for point prediction, anytime-valid admissibility, marginal coverage validity & Cesaro approachability admissibility. The main claims of the authors are that (1) these criteria induce pairwise non-nested admissible classes (2) martingale coherence is not sufficient for Blackwell admissibility under log loss and (3) the 4 criteria can be viewed through a constrained-Bayes lens.

## Key strengths

The paper isolates a recurring ambiguity across several literatures/communities: admissibility depends on the object, order, and certificate. This decomposition gives a sensible and concrete way to compare criteria without treating admissibility as criterion-free.

The Bernoulli vs Bayes example under log-loss is well executed and the authors are able to state a rigorous theorem. It shows that self-consistency or martingale-style calibration does not alone imply Blackwell admissibility.

## Key weaknesses

The paper's main pairwise non-nesting result is stated more strongly than the formal results support. The theorem-level separation is plausible, but much of it follows from the fact that the criteria apply to different procedures and are defined on different partial orders. The manuscript should clarify what important conclusions remain after this structural non-comparability is acknowledged.

I feel that the constrained-Bayes contribution is best presented as a design framework instead of a formal unification. Because the decision spaces / feasible sets / performance criteria differ across settings, the current text slightly overstates the extent of the common optimization structure.

One can certainly argue that ML-oriented discussion goes beyond what the results actually state. The connections to LLM calibration, watermarking, modular AI systems, search agents should probably be described as motivations or analogies rather than as consequences of the theory developed in the paper.

**Audience:**

Yes

**Audience Explanation:**

Researchers working on probabilistic forecasting / sequential inference / conformal prediction / UQ and related topics would likely find the topic relevant and interesting. The criterion-relative framing and the concrete log-loss counterexample provide usable frameworks even if some of the claims in the paper's current format still remain debatable.

**Broader Impact Concerns:**

No specific ethics concern.

**Claims And Evidence:**

Yes

**Claims Explanation:**

The Bernoulli log-loss counterexample is convincing, and the distinction between 4 object spaces and 4 certificates of optimality provides a useful framework for comparing criteria. Overall, the general separation story is plausible.

However, the manuscript states some of its strongest claims too broadly. For example:
- a scope mismatch between the pairwise non-nesting theorems and the broader practical significance sometimes claimed for them
- a scope mismatch in the constrained-Bayes discussion, which supports a general framework but not yet a single unifying formal principle
- application overreach in the ML and industry sections, which are presented more strongly than the theory & experiments support.

The paper should more clearly distinguish between what is formally proved, what depends on particular assumptions, and what is a broader interpretation.

**Requested Changes:**

- In the abstract criterion-separation sentence, Contribution (1), and the Section 10 summary of Theorems 5.9 and 6.7: I think it would make sense to present the result as a witness-based non-nesting theorem across different object spaces and partial orders, rather than as a broader theorem about practical incompatibility.

- In Definition 5.8, Section 5.4, and the Section 10 claim that constrained Bayes **unifies** all four criteria: what about presenting the contribution as a schema or design framework unless additional formal results support a genuinely unified optimization principle.

- In the abstract martingale summary and Contribution (2): I think the authors should distinguish more clearly between what is proved in general v.s. what follows from cited results and what is established by the Bernoulli counterexample.

- In Section 10.1 on industry evidence & similarly in the strongest claims of Section 9.1: I suggest (i.e. it is only a suggestion) moving the discussion of watermarking / modularity / search agents / LLM calibration to motivation, or explicitly present it as interpretation rather than consequence.

---

> ### Author Response · Authors · 2026-06-12
> **Response: scope of claims aligned with what is proved**
>
> Thank you for diagnosing the scope mismatch precisely; we agreed, and all four requested changes are implemented.
>
> (1) The abstract, Contribution (1), and the Section 10 summary now present the result as witness-based non-nesting on a common space: for each ordered pair of the four criterion classes, an explicit predictive system, active in both relevant coordinates, lies in one class but not the other. The text states that the theorem does not assert practical incompatibility, and that any joint ranking requires a normative choice the four theories do not supply.
>
> (2) Constrained Bayes is now a design schema on criterion-specific decision spaces $(\mathcal{D}_C,\mathcal{F}_C,\mathcal{R}_C)$ (Def. 5.16), with Remark 5.17 stating that the four problems are not subproblems of a single optimization: a recipe, not a unifying theorem.
>
> (3) The martingale claims are separated by provenance in the abstract and Contribution (2): proved in general (posterior predictives are martingales under the prior predictive law), cited (AV-admissibility $\Leftrightarrow$ nonnegative martingale, Ramdas et al. 2022), and established by the Bernoulli counterexample (self-consistency does not imply Blackwell admissibility, Thm. 4.4). The sentence suggesting necessity is removed.
>
> (4) Section 9 is retitled "Motivations and interpretations" and opens with a disclaimer that its contents are design interpretations rather than new theorem statements; no watermarking, modular-AI, or search-agent language appears.
>
> The supplementary PDF gives the point-by-point version with locations.

---

> > ### Author Response · Authors · 2026-06-21
> > **Revision 2**
> >
> > We have posted a further revision (rev2) with editorial and correctness refinements only; no theorem, proof, table value, or numerical result has changed. A few items relate to the witness-based, criterion-relative framing you asked us to foreground:
> >
> > - a six-row preview of the separation witnesses now precedes the first separation proof, so the witness-based non-nesting can be read at a glance (an unnumbered display; table numbering is unchanged);
> > - we added a remark (Remark 5.10) relating our cross-criterion separation to the orthogonal, within-$\mathcal{C}_{\mathrm{AV}}$ transport result of Choe & Ramdas (2026, JRSS-B), which situates the contribution in the current anytime-valid literature;
> > - the abstract now opens on the certificate-transport problem, sharpening the framing (its content is unchanged).
> >
> > A full itemization with locations is in Section 5 of the response-to-reviewers PDF. Thank you again for the framing suggestions.

---

### Review · Reviewer_Zuy2 · 2026-06-07

**Summary Of Contributions:**

This is an interesting paper which studies the admissibility geometry of the predictive distribution under a few popular metrics in the literature, including Bayesian and frequentist metrics such as Blackwell admissibility and marginal coverage. The paper shows that the popular criterion geometries are non-overlapping, but attempts to unify them under the lens of a "constrained Bayes" criterion.

**Strengths**: The paper evaluates many popular prediction admissibility criteria under a unified umbrella, which I think is a great idea and will have broad interest. While the results are quite intuitive and expected, the theoretical formalization is appreciated and having it all together in one paper is valuable.

**Weaknesses**: The writing in the paper is unclear in some parts, and quite heavy in jargon. The practical implications are also a little bit understated and delayed until quite late in the paper, making the context of the work a bit confusing at times.

**Audience:**

Yes

**Audience Explanation:**

Yes definitely, as the admissibility criteria spans widely across both Bayesians/frequentsts, as well as both statisticians and machine learners.

**Broader Impact Concerns:**

I have no concerns.

**Claims And Evidence:**

Yes

**Claims Explanation:**

Much of the paper is theoretical, and the results appear reasonable to the best of my knowledge. There are also concrete simulations to illustrate what criterion separation looks like in practice.

**Requested Changes:**

My comments/questions are mostly of minor nature, although I have a few conceptual questions. I think it could strengthen the paper to briefly discuss the below somewhere.

- While the classes are non-nested, it may be valuable to discuss the relative sizes of these classes. I could be mistaken, but it seems intuitive that the Blackwell admissible class is much smaller than the marginal coverage class. I suppose this is also one of the motivations for the constrained Bayes schema, unless I'm missing something.

- In connection with the above, are there any of the classes that are more "compatible"? It would seem that there could be some examples of predictives that satisfy the anytime-valid and coverage criteria.

- Regarding the constrained Bayes approach in Definition 5.8, the prior is treated as fix. However, it is relatively common practice to adjust the prior in order to achieve better frequentist properties. Is there any kind of interpretation here as to what happens when $\Pi$ changes?  For example, does the size of the feasibility set change?

I also have comments regarding a few areas of confusion around the presentation as well as potential typos.
- **Section 2**: The loss function in Definition 2.1 takes an action in the second argument, but then a probability measure in Definition 2.2. Is this a typo? It may also be useful to give an example at the end of this section, e.g. the log loss for the Bernoulli example. Otherwise the notation is a bit dense.

- **Section 3.5**: Is $r*$ corresponding to $\delta*$ in Theorem 3.8? It might be missing a definition. Also, is it necessary to have both the unnormalized $\pi$ and $\Pi$, or can we just stick to the latter?

- **Theorem 5.2**: There is a double reference to Ramdas et al.

- **Figures**: Some of the figures have some formatting issues. The hyperplane in Figure 1 seems to be misaligned, and Figures 2 and 5 are a bit messy with the text overlapping the lines.

---

> ### Author Response · Authors · 2026-06-12
> **Response: all eight requested changes implemented**
>
> Thank you for the encouraging assessment and the constructive suggestions; all eight are implemented.
>
> Relative sizes and compatibility: a new Section 5.7 discusses the geometric size of each class (without formal cardinality claims) and exhibits three explicit intersections, including $\mathfrak{A}\cap\mathfrak{C}$ (likelihood-ratio $e$-process paired with a split-conformal set) and $\mathfrak{B}\cap\mathfrak{D}$ on a compact Bernoulli submodel; non-nesting forbids universal subsumption, not coexistence.
>
> Prior sensitivity: Remark 5.18 distinguishes the regime where $\Pi$ enters only the objective (feasibility unchanged; $\Pi$ rotates the supporting hyperplane of the restricted risk set) from the regime where $\Pi$ also enters the feasible set; the paper's results sit in the first.
>
> Notation and presentation: Definition 2.3 writes the risk as an explicit marginalization, resolving the action-vs-measure appearance; Example 2.4 introduces the Bernoulli log-loss running example at the end of Section 2; a notation paragraph at the head of Section 3.5 fixes the $\pi$ vs. $\Pi$ convention ($\pi$ geometric, $\Pi=\pi/\|\pi\|_1$ probabilistic); the duplicate Ramdas et al. citation is removed.
>
> Figures: all redrawn and re-checked at high magnification. The Figure 1 hyperplane now passes exactly through $r^*$ with the normal perpendicular and correctly oriented; Figure 2's two supporting hyperplanes are genuinely tangent at their Bayes points with prior labels moved clear of the curves; Figure 5 is resized with labels on white backgrounds clear of the dashed arrows.
>
> The supplementary PDF contains the point-by-point response with manuscript locations.

---

> > ### Author Response · Authors · 2026-06-21
> > **Revision 2.**
> >
> > We have posted a further revision (rev2) with editorial and correctness refinements only; no theorem, proof, table value, or numerical result has changed. Several of these touch the presentation and notation you helped us tighten in the previous round:
> >
> > - probability notation is now uniform throughout -- $\mathbb{P}$ for the coverage probability and the model law $P_\theta$ for event probabilities (the roman "Pr" was removed);
> > - we repaired four cross-references that had rendered as a phantom "Remark 3" (the label sat on an unnumbered paragraph);
> > - the acronym "CApp" is standardized (the variant "CAA" was removed), a few rhetorical metaphors were trimmed, and spelling was made consistent;
> > - the abstract's opening was reframed around the certificate-transport problem (its content -- the four geometries, the space $\Sigma$, and the witness-based theorem -- is unchanged).
> >
> > A full itemization with locations is in Section 5 of the response-to-reviewers PDF. Thank you again for the notation and presentation suggestions.